# Stage-specific GATA3 induction promotes ILC2 development after lineage commitment

Hiroki Furuya [1,5], Yosuke Toda [1,5], Arifumi Iwata [1,5] ✉, Mizuki Kanai[1], Kodai Kato[1], Takashi Kumagai[1], Takahiro Kageyama [1], Shigeru Tanaka [1], Lisa Fujimura[2], Akemi Sakamoto[2,3], Masahiko Hatano[2,3], Akira Suto [1], Kotaro Suzuki [1] & Hiroshi Nakajima [1,4] ✉

Group 2 innate lymphoid cells (ILC2s) are a subset of innate lymphocytes that produce type 2 cytokines, including IL-4, IL-5, and IL-13. GATA3 is a critical transcription factor for ILC2 development at multiple stages. However, when and how GATA3 is induced to the levels required for ILC2 development remains unclear. Herein, we identify ILC2-specific GATA3-related tandem super-enhancers (G3SE) that induce high GATA3 in ILC2-committed precursors. G3SE-deficient mice exhibit ILC2 deficiency in the bone marrow, lung, liver, and small intestine with minimal impact on other ILC lineages or Th2 cells. Single-cell RNA-sequencing and subsequent flow cytometry analysis show that GATA3 induction mechanism, which is required for entering the ILC2 stage, is lost in IL-17RB+PD-1− late ILC2-committed precursor stage in G3SE-deficient mice. Cnot6l, part of the CCR4-NOT deadenylase complex, is a possible GATA3 target during ILC2 development. Our findings implicate a stage-specific regulatory mechanism for GATA3 expression during ILC2 development.

Group 2 innate lymphoid cells (ILC2s) are a subset of innate lymphocytes that lack antigen receptors but are functionally similar to type 2 helper T (Th2) cells and produce type 2 cytokines, such as IL-4, IL-5, and IL-13. As a result, ILC2s play pivotal roles in the pathophysiology of bronchial asthma, chronic rhinosinusitis, atopic dermatitis, and organ fibrosis[1–5]. Whereas Th2 cell differentiation requires IL-4/STAT6 signaling, ILC2 development is independent of IL-4/STAT6, and there is no available therapy specifically targeting ILC2s[6,7].

ILC-lineage cells develop from ILC progenitors (ILCPs) that express PLZF (encoded by *Zbtb16*), PD-1, and LPAM-1 (Integrin α4β7), but no ST2 or CD25 in bone marrow (BM)[8–10]. ILCPs have the potential to develop into ILC1s, ILC2s, and ILC3s but not into NK cells or lymphoid tissue inducer (LTi) cells. Within the ILCP population, ILC2-committed precursors were identified using the expression of ICOS[11], *Bcl11b*[10,12,13], and IL17RB[10] along with PD-1 and PLZF. The stage between

*Zbtb16+* cells, including ILC2-committed precursors, and ILC2s was recently identified. Kasal et al. reported that Lin−CD127+LPAM-1+*Id2*+CD25−CD135−ICOS+*Zbtb16*− cells could develop only into ILC2s[14]. Hence, these cells were considered as the next stage of *Zbtb16*+ ILC2-committed precursors. Finally, ILC2s in BM express ST2 and CD25, serving as the source of tissue ILC2s[8–11,15] and playing unique roles in BM under the stress conditions[16,17]. In addition, ILCPs were recently identified in lung tissues where they contribute to maintaining ILC2 homeostasis under normal physiological conditions or ILC2 induction during inflammation[8,18–20]. However, how the final-stage ILC2s are derived from the ILC2-committed precursors in the BM and ILCPs in the lung remains largely unknown.

GATA3 is a critical transcription factor for early ILC development and ILC2 development, as well as early T cell development and Th2 differentiation at multiple stages, acting in a dose-dependent

[1]Department of Allergy and Clinical Immunology, Graduate School of Medicine, Chiba University, Chiba, Japan. [2]Biomedical Research Center, Chiba University, Chiba, Japan. [3]Department of Biomedical Science, Graduate School of Medicine, Chiba University, Chiba, Japan. [4]Chiba University Synergy Institute for Futuristic Mucosal Vaccine Research and Development (cSIMVa), Chiba, Japan. [5]These authors contributed equally: Hiroki Furuya, Yosuke Toda, Arifumi Iwata. ✉e-mail: aiwata@chiba-u.jp; nakajimh@faculty.chiba-u.jp

manner[3,21–24]. GATA3 is essential for developing early innate lymphoid progenitors, ILCPs, and maintaining mature ILC2s[15,22,25,26]. A recent study demonstrated the involvement of a GATA3 enhancer region that contributes to ILC2 development[27]. However, when and how GATA3 is induced up to the appropriate levels required for ILC2 development remains unclear.

Herein, we identify that high-GATA3 expression essential for ILC2 development is regulated by ILC2-specific tandem GATA3-related super-enhancers (G3SE), which is required specifically in IL17RB⁺PD-1⁻ late ILC2-committed precursors for entering the ILC2 stage. Our findings show a stage-specific regulatory mechanism and roles of GATA3 expression during ILC2 development, which can lead to a new therapy specifically targeting ILC2s.

## Results

### GATA3-related tandem super-enhancers are indispensable for the development of peripheral ILC2s

Super-enhancers (SEs) are regions composed of clusters of active enhancers, which are known to drive the expression of genes that define cell identity and function[28,29]. To identify ILC2-specific SEs, we performed H3K27ac chromatin immunoprecipitation sequencing (ChIP-seq) of lung ILC2s, defined as Lin⁻CD45⁺Thy1⁺ST2⁺ cells, and lung CD4 T cells in house dust mite-induced asthma models, which recapitulate activated ILC2s and activated Th2 cells in vivo, respectively (Supplementary Fig. 1a, b). We identified 624 SEs (Supplementary Fig. 1c, d) and found that most were similarly activated in ILC2s and CD4 T cells, as previously reported[30]. For example, SE-*Gata3*, which includes the *Gata3* gene body, and the type 2 cytokine locus SE-*Rad50* exhibited similar activation between ILC2s and CD4 T cells under asthmatic conditions (Fig. 1a and Supplementary Fig 1c, e, f).

In the SEs which were specifically activated in ILC2 (fold change >4; Supplementary Fig. 1e, n = 36), two tandem SEs were located at 678–764 and 500–591 kb downstream from the *Gata3* transcription start site (TSS), respectively (SE1 and SE2, GATA3-related tandem super-enhancers; G3SE). SE1 and SE2 were already activated in ILC2s under control conditions, exhibiting lower activation in CD4 T cells, as was also the case under asthmatic conditions (Fig. 1a). Public ATAC-seq (assay for transposase-accessible chromatin with high-throughput sequencing) data for helper T cell subsets[31] and our ATAC-seq data for lung ILC2s under steady-state conditions demonstrated that both SE1 and SE2 were accessible in lung ILC2s but were less accessible in naïve, Th1, and Th2 cells (Supplementary Fig. 1g). These data indicate that G3SE might play an important role in GATA3 expression in lung ILC2s.

To investigate the function of G3SE, we generated G3SE-knockout (G3SEKO) mice using the CRISPR-Cas9 system with two guide RNAs flanking the entire region of the tandem SEs (Supplementary Fig. 2a). We successfully obtained two lines, 37 bp deletion and 11 bp deletion (Supplementary Fig. 2b), which have no obvious off-target effect (Supplementary Fig. 3). Importantly, G3SEKO mice exhibited reduced lung ILCs compared to wild-type (WT) mice (Fig. 1b and Supplementary Fig. 4a), and the remaining ILCs expressed GATA3 at low levels (Fig. 1c). Consequently, G3SEKO mice showed significantly reduced lung ILC2s, based not only on GATA3 or ST2, but also on KLRG1, an ILC2 marker independent of GATA3[26] (Fig. 1c and Supplementary Fig. 4b). T-bet⁺ ILC1s tended to increase, while RORγt⁺ ILC3s were not affected, in G3SEKO mice (Fig. 1c and Supplementary Fig. 4b). We also confirmed ILC2 deficiency in other peripheral organs, such as the liver and small intestine, in G3SEKO mice (Fig. 1d, e). ILC1s tended to increase in the liver and significantly increased in the small intestine of G3SEKO mice (Fig. 1d, e and Supplementary Fig. 4c, d). However, other immune cell populations in the lung and liver (Supplementary Fig. 5a, b) and T cell development and GATA3 expression in the thymus (Supplementary Fig. 5c, d) were indistinguishable between G3SEKO and WT mice. Intriguingly, the expression of helper T cell subset

master regulators, including GATA3, was not affected in G3SEKO mice (Supplementary Fig. 5e).

Regarding Th2 cells, it has recently been reported that the SE1 region is opened in *N. brasiliensis*-activated lung Th2 cells[27,30]. Because CD4 T cells obtained from the lung in HDM-induced asthma models are not exclusively Th2 cells, the low activation of G3SE in lung CD4 T cells (Fig. 1a) might result from the dilution of Th2 cells by other CD4 T cell subsets. Therefore, we assessed in vivo Th2 cell development and function using ovalbumin (OVA)-induced asthma models (Supplementary Fig. 6a), which are known to be less influenced by ILC2 deficiency. Notably, the numbers of eosinophils, Th2 cells (defined by CD3ε⁺CD4⁺ST2⁺ T cells), and IL-5- or IL-13-producing Th2 cells were comparable between WT and G3SEKO mice (Supplementary Fig. 6b–e), while GATA3 expression in Th2 cells in lung and bronchoalveolar lavage fluid (BALF) was slightly decreased in G3SEKO mice (Supplementary Fig. 6f). These results suggest that the G3SE region is critical in ILC2 development but only partially impacts Th2 cell development.

Next, we conducted mixed BM chimera analysis to determine whether the reduction in ILC2s is cell-intrinsic (Fig. 1f). The ratio of ILC2s derived from G3SEKO BM cells to ILC2s derived from WT BM cells was considerably low in the lung, liver, and small intestine, whereas that of ILC1s derived from G3SEKO BM cells was moderately high in the liver and small intestine (Fig. 1g). The ratio of ILC3 in the lung, liver, and small intestine was not affected by the absence of G3SE (Fig. 1g). These results suggest that G3SE promotes the development of ILC2s in a cell-intrinsic manner.

### High-GATA3 expression induced by G3SE is essential for the development from ILCP to ILC2

In adult mice, ILC2s are not only self-renewed in peripheral tissues but also developed from CLP in BM. To investigate the specific developmental stage regulated by G3SE, we first explain the terms and definition of the ILC2 development stage used in this study (Supplementary Fig. 7a) because, in the BM, ILC2 cells have been referred to by various names such as LSIG[15], ILC2P[13–15,22], iILC2[9], and BM ILC2[12,16,17]. In this study, we defined ILC2s in the BM as PD-1⁻ST2⁺ cells, terming them ST2⁺ ILC2s. In addition, cells already committed to ILC2 lineages but not yet exhibiting the ST2-positive stage (named as ILC2-committed precursors) have been recently identified as *Bcl11b*⁺IL17RB⁺PD-1⁺ cells[10,12] or ICOS^hi*Zbtb16*⁺ cells[11] among PD-1⁺ ILCPs. CD25⁻*Bcl11b*⁺*Zbtb16*⁻ cells and CD25⁻ICOS^hi*Zbtb16*⁻ cells were also identified as ILC2-committed precursors[12,14]. In this study, we distinguished ILC2-committed precursors by PD-1 expression: namely, we termed PD-1⁺ cells[10–12] closer to ILCPs as early ILC2-committed precursors and PD-1⁻ cells[12,14] closer to ILC2s as late ILC2-committed precursors. Moreover, ILCPs were defined as PD-1⁺ST2⁻IL17RB⁻ cells and referred to as IL17RB⁻ ILCPs for excluding early ILC2-committed precursors (Supplementary Fig. 7a).

We then examined the differentiation stages of BM cells according to the classification above. We first analyzed α-lymphoid progenitor (αLP) population (Lin⁻CD127⁺CD135⁻LPAM-1⁺ cells), which encompasses IL17RB⁻ ILCPs, early and late ILC2-committed precursors, ST2⁺ ILC2s, and PD-1⁻ST2⁻ cells (also known as CHILPs[32], but now recognized as the mixture of iLTiP and iILCP[11,14,22]). Notably, late ILC2-committed precursors were scarcely observed within αLP when ST2 was used as an ILC2 marker (Supplementary Fig. 7b), which is a contrast to the findings using CD25 as an ILC2 marker[14]. The discrepancy can be attributed to using ST2 as an ILC2 marker instead of CD25 because approximately 5% of ST2⁺ ILC2s are CD25-negative (Supplementary Fig. 7c). Previous studies showed that GATA3 deficiency in hematopoietic cells diminishes PD1⁻ST2⁻ cells and completely depletes PD-1⁺ ILCPs[25,26,33]. Importantly, in G3SEKO mice, there was no significant impairment in the development of IL17RB⁻ ILCPs and early ILC2-committed precursors; however, ST2⁺ ILC2s were almost completely absent (Fig. 2a).

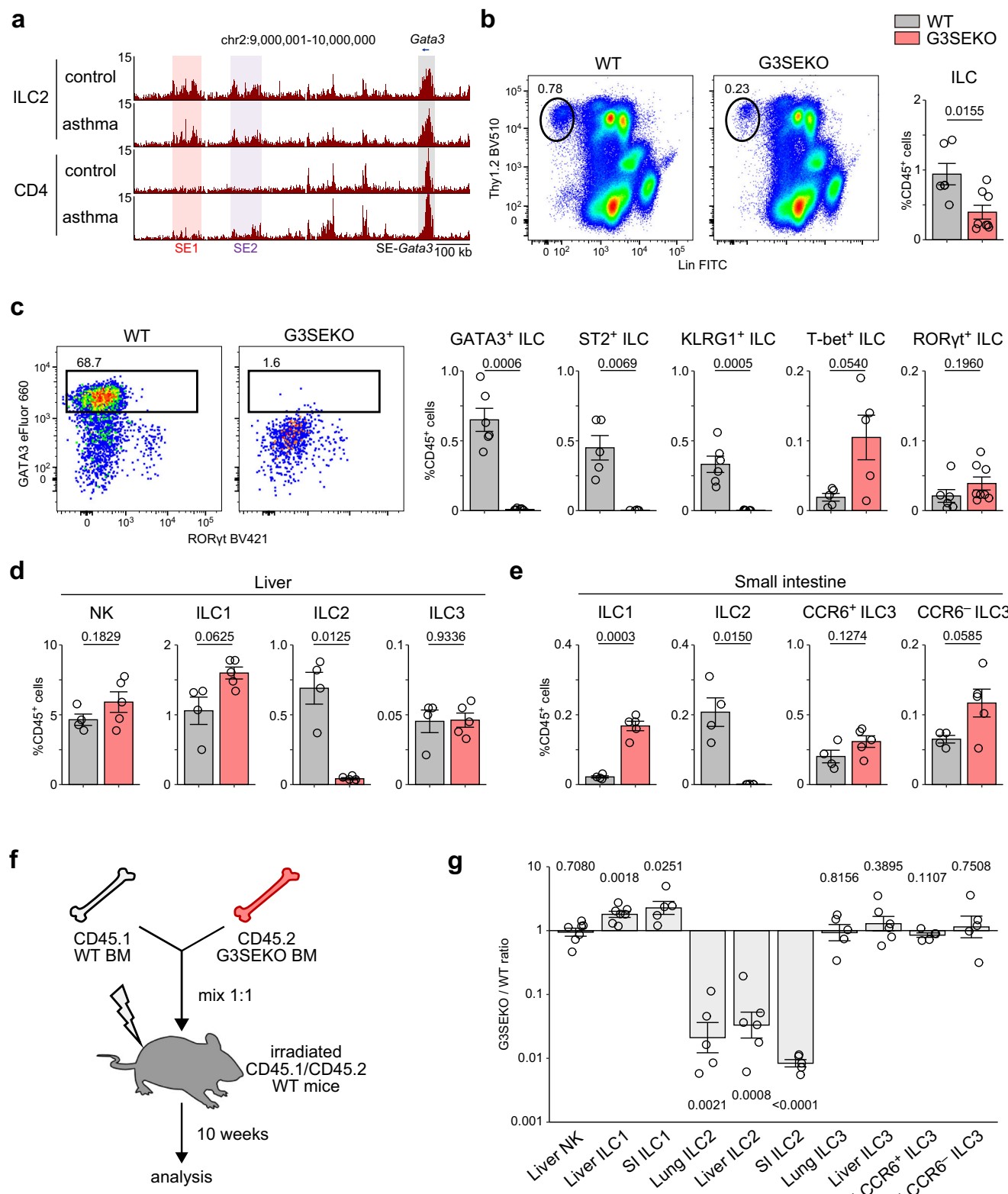

To ascertain whether G3SE regulates ILC2 development through GATA3 induction, we tested if forced GATA3 expression in G3SEKO BM cells could rescue ILC2 development. First, we performed an adoptive transfer of GATA3-overexpressing G3SEKO BM cells into irradiated WT mice. However, those cells did not differentiate into lymphoid cells, consistent with a previous report[34]. Therefore, we performed in vitro ILCP culture with OP9-DL1 cells, focusing on the development of ILC2s from ILCPs. In this culture

system, whereas WT ILCPs successfully lost PD-1 expression and differentiated into CD25+PD-1− cells, G3SEKO ILCPs did not differentiate into CD25+PD-1− cells, and substantial numbers of CD25−PD-1+ cells remained (Fig. 2b). Furthermore, no GATA3high cells developed from G3SEKO ILCPs (Fig. 2c). Importantly, the overexpression of GATA3 rescued the development of CD25+PD-1− cells from G3SEKO ILCPs, accompanied by a reduction in the remaining CD25−PD-1+ cells (Fig. 2d). These data suggest that high-GATA3

**Fig. 1 | GATA3-related tandem super-enhancers are indispensable for the development of peripheral ILC2s. a** ChIP-seq analysis of H3K27 acetylation in ILC2s and CD4 T cells in lungs from control and asthmatic mice. Rectangles indicate super-enhancers. See also Supplementary Fig. 1a–e. **b** ILC frequency (CD45$^+$Lin$^-$Thy1.2$^+$ cells) in lung CD45$^+$ cells of WT and G3SEKO mice. The gating strategy is shown in Supplementary Fig. 4a, WT: n = 6, G3SEKO: n = 8. **c** Frequency of ILC2s defined by different markers (GATA3$^+$, ST2$^+$, or KLRG1$^+$), ILC1s (T-bet$^+$), and ILC3 (RORγt$^+$) in lung CD45$^+$ cells. See also Supplementary Fig. 4b. ST2$^+$ ILC, T-bet$^+$ ILC: n = 5, GATA3$^+$ ILC, KLRG1$^+$ ILC, RORγt$^+$ ILC, WT: n = 6, G3SEKO: n = 8. **d** NK, ILC1, ILC2, and ILC3 frequency in liver CD45$^+$ cells. The gating strategy is shown in Supplementary Fig. 4c. WT: n = 4, G3SEKO: n = 5. **e** ILC1, ILC2, and ILC3 frequencies

in small intestine CD45$^+$ cells. The gating strategy is shown in Supplementary Fig. 4d, WT: n = 4, G3SEKO: n = 5. **f** Scheme of mixed bone marrow (BM) chimeric mouse experiment. **g** Ratio of CD45.2$^+$ G3SEKO/CD45.1$^+$ WT cells in the indicated cell population in mixed BM chimeric mice. Liver NK, ILC1, n = 7, Liver ILC2, ILC3: n = 6, SI ILC1, ILC2, CCR6$^+$/CCR6$^-$ ILC3, Lung ILC2, ILC3: n = 5. **b–e, g** Each data point indicates one mouse from four (**b, c**) or two (**d, e, g**) independent experiments. Data are presented as mean ± SE. Statistical analysis was performed using unpaired, two-sided Welch's $t$-test (**b–e**) or the one-sample, two-sided Welch's $t$-test to a ratio=1 (**g**). p values are shown on the graphs. Source data are provided as a Source Data file.

expression induced by G3SE is essential to drive ILC2 development after the ILCP stage.

## ILC2 development after ILC2-lineage commitment is dependent on GATA3 induction by G3SE

To further investigate the precise stage of ILC2 development that requires high-GATA3 expression induced by G3SE, we performed single-cell RNA sequencing (scRNA-seq) for αLPs in WT and G3SEKO mice (Supplementary Fig. 8a). The cells with low *Id2* expression (ILC lineage marker[12]) were excluded from the analyses. Consequently, 654 αLPs from WT BM and 39 αLPs from G3SEKO BM in the 1st experiment and 5230 WT αLPs and 1140 G3SEKO αLPs in the 2nd experiment were integrated and subjected to further analyses. Cells were classified into 8 clusters (Fig. 3a), and the cells in all clusters expressed *Id2* and *Il7r*, indicating that these clusters belong to ILC lineages (Fig. 3b). To validate the cell populations, the marker genes during ILC lineage development were analyzed in WT cells. Cluster 0 expressed high levels of *Rorc* and *Ccr6*, suggesting that these cells are LTi cells/ILC3s (Supplementary Fig. 8b). Cluster 1 expressed high levels of *Tbx21* (T-bet), *Klrb1c* (NK1.1), and *Il2rb*, suggesting that these cells are NK cells/ILC1s[10,12,35] (Fig. 3b, c, Supplementary Fig. 8b). Expression levels of ILCP markers *Zbtb16* (PLZF), *Pdcd1* (PD-1), *Tox*, and *Tox2* were dominantly upregulated in clusters 2 and 3. The cells in clusters 3 and 4 expressed early ILC2 markers, such as *Icos*, *Bcl11b*, and *Il17rb* but not mature ILC2 markers *Il1rl1* (ST2) and *Il2ra* (CD25) (Fig. 3b, c, Supplementary Fig. 8b, c), suggesting that clusters 2, 3, and 4 are IL17RB$^-$ ILCPs, early ILC2-committed precursors, and late ILC2-committed precursors, respectively. Cells in cluster 5 expressed *Mki67* and *Top2a*, suggesting that these cells are proliferating cells. Part of cells in clusters 5 and 6 started to express the ILC2 markers *Il1rl1* and *Il2ra*, and most of the cells in cluster 7 expressed both *Il1rl1* and *Il2ra*, suggesting that cluster 7 were ST2$^+$ ILC2s, and clusters 5 and 6 were a mixture of late ILC2-committed precursors and ST2$^+$ ILC2s (Fig. 3b, c, Supplementary Fig. 8b, c). Importantly, G3SEKO cells developed into cluster 6 but not in cluster 7 (Fig. 3e, f). This finding was consistent with *Gata3* expression levels, where *Gata3* expression in G3SEKO cells began to decrease in cluster 3 and markedly decreased in clusters 5 and 6 compared to WT cells (Fig. 3g). Notably, G3SEKO cells in clusters 5 and 6 exhibited prolonged expression of some feature genes of clusters 2 and 3, such as *Il18r1*, *Atp8b*, *Ikzf3*, and *Pecam1* (Fig. 3h). Intriguingly, *Il18r1* is highly expressed in the ILCP, early and late ILC2-committed precursors, but not in the mature ILC2s in WT cells (cluster 7), indicating that *Il18r1* could be a helpful marker of late ILC2-committed precursors[36]. These data indicate that G3SEKO cells develop into the late ILC2-committed precursor stage but not the ST2$^+$ ILC2 stage.

To trace the development of ILC2 lineages, we questioned the use of LPAM-1$^+$ gating for ILC2 lineage-tracing because of the presence of LPAM-1$^-$ST2$^+$ ILC2s[15] (Supplementary Fig. 9a). IL17RB$^-$ ILCPs expressed high levels of LPAM-1, and their expression reduced along with IL17RB induction (Supplementary Fig. 9a, b). ST2$^-$IL17RB$^+$ cells, representing putative early and late ILC2-committed precursors, sustained low levels of LPAM-1 during PD-1 downregulation (Supplementary Fig. 9c). To examine the differentiation potential of IL17RB$^+$PD-1$^-$ cells, we

cultured these cells on OP9-DL1 cells. Five days later, IL17RB$^+$PD-1$^-$ cells developed mostly into CD25$^+$ICOS$^{hi}$ ILC2s (Supplementary Fig. 9d). Collectively, we defined late ILC2-committed precursors as Lin$^-$CD127$^+$CD135$^-$ST2$^-$IL18Rα$^+$PD-1$^-$IL17RB$^+$ cells. This observation aligns with the results obtained from scRNA-seq analysis.

In G3SEKO mice, IL17RB$^-$ ILCP and early ILC2-committed precursors were comparable to those in WT mice (Fig. 3i). However, following PD-1 downregulation, late ILC2-committed precursors were significantly accumulated (Fig. 3i). In both WT and G3SEKO mice, late ILC2-committed precursors did not express T-bet or RORγt (Supplementary Fig. 9e), suggesting that the accumulated late ILC2-committed precursors are not ILC1 or ILC3 lineage-committed cells.

Next, we examined the expression levels of GATA3 protein among these cells. In WT mice, GATA3 was highly expressed in IL17RB$^-$ ILCPs, slightly upregulated in early ILC2-committed precursors, and gradually reduced in late ILC2-committed precursors (Fig. 3j). GATA3 expression in IL17RB$^-$ ILCPs in G3SEKO mice was comparable to that in WT mice. However, GATA3 expression was not upregulated in early ILC2-committed precursors and significantly reduced in late ILC2-committed precursors in G3SEKO mice (Fig. 3j).

To understand the activation status of SE1 and SE2 during ILC2-lineage development in the BM, we performed ATAC-seq for IL17RB$^-$ ILCP, early and late ILC2-committed precursors, IL18Rα$^+$ST2$^+$ ILC2s, and IL18Rα$^-$ST2$^+$ ILC2s. Both SE1 and SE2 contained opened chromatin regions in late ILC2-committed precursors, which are already opened in early ILC2-committed precursors but not in IL17RB$^-$PD-1$^+$ ILCPs (Supplementary Fig. 10, purple triangles). A substantial number of open chromatin regions in IL17RB$^-$ ILCPs were closed in late ILC2-committed precursors (Supplementary Fig. 10, blue triangles). These data suggest that the G3SE starts to function in early ILC2-committed precursors, collaborating with unknown ILCP-specific GATA3 induction mechanisms. Further, when cells enter the late ILC2-committed precursor stage, the ILCP-specific GATA3 induction mechanism is lost, and GATA3 expression largely depends on G3SE activation for entering the ILC2 stage.

## High-GATA3 expression in lung ILCP-like cells does not depend on G3SE

Recently, two groups reported that a fraction of lung ILC2s is derived from lung ST2$^-$IL18Rα$^+$ ILCPs[18,20]. For instance, Zeis et al. have shown that a small portion of cells in the lung ST2$^-$IL18Rα$^+$ fraction expresses PLZF, PD-1, and high levels of GATA3, like bone marrow ILCPs[20]. To evaluate the role of GATA3 regulated by G3SE in ILC2 development from ILCPs in the lung, we first identified ILCP-like cells using PLZF and PD-1 as markers. Importantly, PLZF$^{high}$PD-1$^{high}$ ILCP-like cells were observed in the lungs in both WT and G3SEKO mice (Fig. 4a). The ILCP-like cells expressed significant levels of GATA3 but did not express ST2 in both mice (Fig. 4b). Surprisingly, most cells with high levels of GATA3 expression in the G3SEKO lung were ILCP-like cells (Fig. 4b). These data indicate that the GATA3 induction mechanism in ILCP-like cells in the lung does not depend on G3SE.

It has also been shown that lung ST2$^-$IL18Rα$^+$IL17RB$^+$ cells mainly give rise to ILC2s in the culture, in contrast to ST2$^-$IL18Rα$^+$IL17RB$^-$

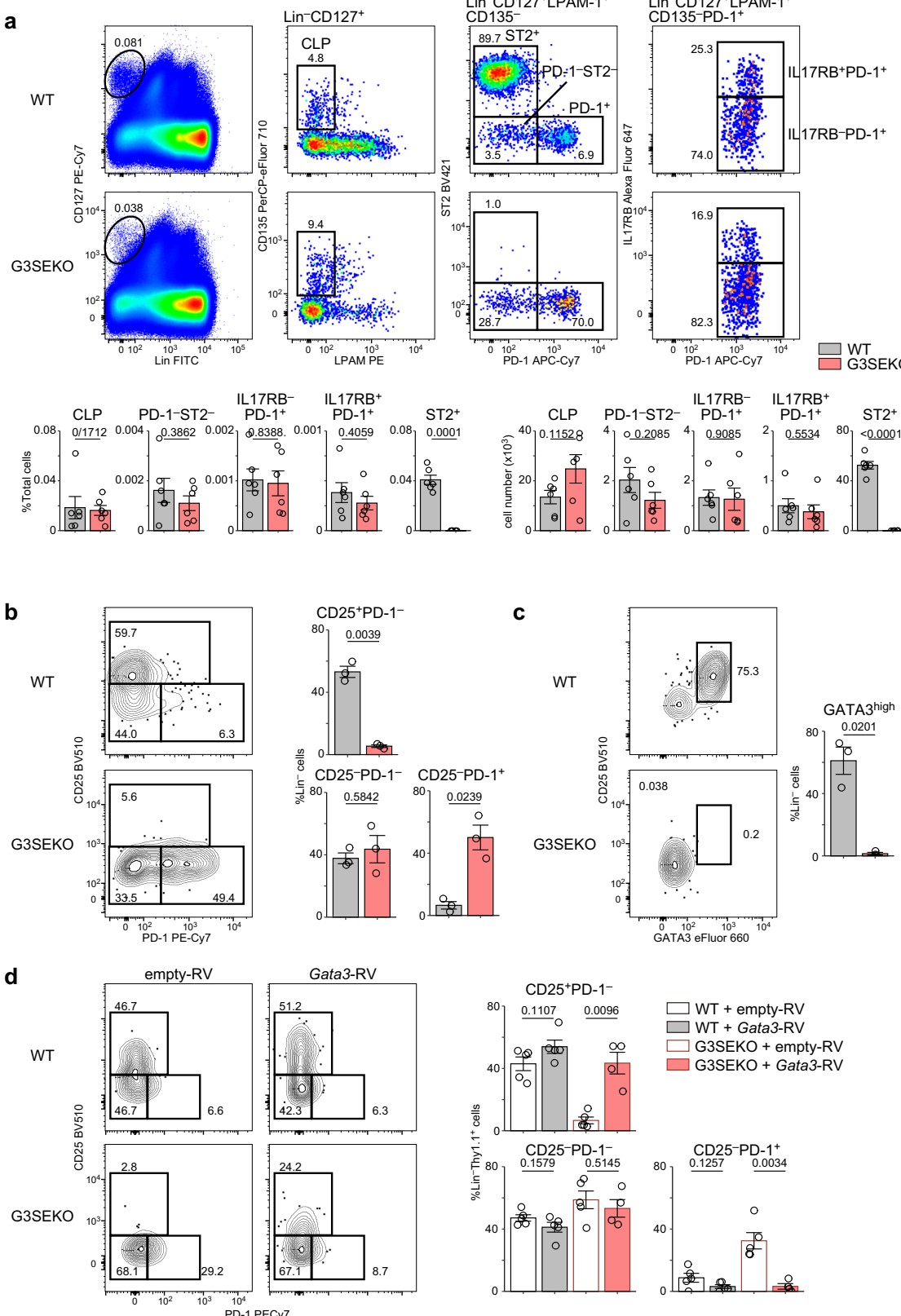

cells[20]. We, therefore, examined ILC2-committed precursors in the lung using IL17RB, applying the staining strategies used in the BM. Both IL17RB−PD-1+ cells and IL17RB+PD-1+ cells were comparable between WT and G3SEKO mice; however, IL17RB+PD-1− cells, presumably late ILC2-committed precursors, were significantly increased in G3SEKO mice by approximately 6-fold (Fig. 4c). Another ILC2 marker, Sca-1[15,26], started

to express in ST2−IL18Rα+ cells, with subsequent upregulation during ILC2 development in WT mice (Supplementary Fig. 11a). Notably, in G3SEKO mice, the majority of accumulated ST2−IL18Rα+ cells expressed Sca-1 (Supplementary Fig. 11b). These results suggest that G3SE promotes the transition from IL17RB+PD-1− cells to ST2+ ILC2s in the lung.

**Fig. 2 | High-GATA3 expression induced by G3SE is essential for the development from ILCP to ILC2. a** Flow cytometry analysis of BM cells. Frequencies and numbers of CLPs (Lin⁻CD127⁺CD135⁺LPAM-1⁻ cells), PD-1⁻ST2⁻ (Lin⁻CD127⁺CD135⁻LPAM-1⁺PD-1⁻ST2⁻ cells), IL17RB⁻PD-1⁺ (Lin⁻CD127⁺CD135⁻LPAM-1⁺PD-1⁺ST2⁻IL17RB⁻ cells), IL17RB⁺PD-1⁺ (Lin⁻CD127⁺CD135⁻LPAM-1⁺PD-1⁺ST2⁻IL17RB⁺ cells), and ST2⁺ ILC2s (Lin⁻CD127⁺CD135⁻LPAM-1⁺PD-1⁻ST2⁺ cells) were evaluated; n = 6. Lin: CD3ε, CD4, CD5, CD8a, CD11b, CD11c, CD19, NK1.1, Gr1, B220, Ter119. **b**, **c** Sorted ILCPs from the BM of WT mice and G3SEKO mice were cultured on OP9-DL1 cells in the presence of IL-7 and SCF for 5 days. Frequencies of Lin⁻CD25⁺PD-1⁻ cells, Lin⁻CD25⁻PD-1⁻ cells, Lin⁻CD25⁻PD-1⁺ cells (**b**), and Lin⁻CD25⁺GATA3⁺ cells (**c**) were evaluated, n = 3. **d** ILCPs sorted from the BM of WT mice and G3SEKO mice were cultured on OP9-DL1 cells. Cells were infected with GATA3-expressing or empty retrovirus on day 1. The infected cells (Thy1.1⁺ cells) were analyzed on day 6. WT, G3SEKO+empty-RV: n = 5, G3SEKO+Gata3-RV: n = 4. Each data point indicates one mouse from six (**a**) or three (**b**–**d**) independent experiments. Data are presented as mean ± SE. Statistical analysis was performed using unpaired, two-sided Welch's $t$-test. p values are shown on the graphs. Source data are provided as a Source Data file.

### *Cnot6l* is a possible target of GATA3 during ILC2 development

To identify genes with critical roles in ILC2 development from ILC2-committed precursors, which are directly regulated by high levels of GATA3 through G3SE, we screened genes using three criteria: (1) genes upregulated during ILC2 development; (2) differentially expressed genes (DEGs) between WT and G3SEKO cells; (3) genes with a GATA3-binding site and a chromatin state opened from ILCPs to ILC2s by G3SE. To obtain genes upregulated during ILC2 development, we first employed tradeSeq to analyze the kinetics of expression for genes exhibiting variable expression from cluster 2 to cluster 7 and then categorized the genes into four unbiased groups based on pseudotime analysis (Fig. 5a and Supplementary Fig. 12a). Expression levels of Group I genes were upregulated toward the end of ILC2 development, those of Group II genes were upregulated in ILC2-committed precursors, those of Group III and Group IV genes were upregulated in ILCPs but downregulated throughout ILC2 development (Fig. 5b).

Second, we analyzed DEGs between WT and G3SEKO mice in each cluster. There were relatively low numbers of DEGs in clusters 2−6 (Supplementary Fig. 12b). In contrast, cluster 7, consistent with impaired ILC2 differentiation in G3SEKO mice, harbored a significant number of DEGs between WT and G3SEKO mice (Fig. 5c). Upregulated genes in cluster 7 in WT mice were significantly enriched for genes categorized as Group I (Fig. 5d). In contrast, upregulated genes in cluster 7 in G3SEKO mice were significantly enriched for genes categorized as Group II and III (Fig. 5d). Meanwhile, there was no significant enrichment of genes from Group IV. These findings indicate that cells in cluster 7 in G3SEKO mice still exhibit the transcriptional profile of ILCPs and ILC2-committed precursors rather than ILC2s.

Third, we searched genes with GATA3-binding sites in open chromatin regions during differentiation from ILCPs to ILC2s. ATAC-seq data of WT ST2⁺IL18Rα⁻ ILC2 and WT IL17RB⁻ ILCP were used to identify open chromatin regions, ATAC-seq data of WT ST2⁺IL18Rα⁻ ILC2 and G3SEKO late ILC2-committed precursors were used to identify G3SE-mediated GATA3-dependent regions, and published GATA3 ChIP-seq data of cultured ILC2[37] were used to identify GATA3-binding peaks. We found 362 genes with open chromatin regions overlapping GATA3-binding peaks in the gene body or within 20 kb from the TSS. For example, *Il1rl1* had two regions with GATA3-binding and open chromatin regions, and *Cnot6l* had one region (Fig. 5e). In contrast, *Zfp36l1*, which was strongly upregulated in cluster 7 in WT but not in G3SEKO mice, had several newly opened chromatin regions in ILC2s yet without GATA3 binding, indicating that GATA3 induces *Zfp36l1* expression in an indirect manner (Fig. 5c, e).

By integrating these criteria, we identified five candidate genes: *Il1rl1, Slc7a8, Cnot6l, Ptpn13*, and *Inpp4b* (Fig. 5f and Supplementary Fig. 12c). *Il1rl1*, encoding ST2, is a most useful marker of ILC2s that is strongly dependent on GATA3, and ST2-deficient mice exhibit normal ILC2 development in the BM but have a defect in peripheral migration[26,38]. *Slc7a8* is a transporter of arginine and large amino acids. *Il7ra*ᶜʳᵉ*Slc7a8*ᶠˡ/ᶠˡ mice have a reduced number of ILC2, and the rest of the ILC2s fail to produce IL-5 and IL-13[39]. *Cnot6l* is one of the components of the CCR4-NOT complex, which functions as an mRNA deadenylase,

and the CCR4-NOT complex is critical in early T cell development[40]. *Ptpn13* is a tyrosine phosphatase that interacts with various molecules and regulates Th1 and Th2 differentiation through STAT4 signaling[41]. *Inpp4b* is a signaling protein that regulates the PI3K/Akt pathway through phosphatase activity, which hydrolyzes PI(3,4,5)P3 and PI(3,4) P2. It has recently been shown that *Inpp4b* directs ILC1 homing to cancer tissues using NCRᶦᶜʳᵉ/*Inpp4b*ᶠˡ/ᶠˡ mice[42]. Because two of the five candidate genes, *Il1rl1* and *Slc7a8*, are reported as critical molecules for ILC2 development and functions, we hypothesize that the remaining three genes may also be crucial for ILC2 development.

We, therefore, performed in vivo knockdown experiments on these genes using shRNA retrovirus systems (Fig. 5g). WT BM cells were transfected with knockdown retroviruses and transferred into irradiated congenic mice. Five weeks later, lung and BM cells derived from the transduced cells were analyzed. The proportion of ST2⁺ ILC2s was significantly decreased in the lung and BM cells derived from *Cnot6l*-targeted shRNA-transduced cells without affecting GATA3 expression (Fig. 5h, i, Supplementary Fig. 12d). BM cells transduced with shRNA targeting *Inpp4b* or *Ptpn13* normally developed into ILC2s (Supplementary Fig. 12e). The *Cnot6l*-enhancer region, identified by ATAC-seq (Fig. 5e), was accessible during ILC2 development in WT mice but maintained a closed conformation in G3SEKO ILCs (Fig. 5e). We also conducted a luciferase assay targeting the *Cnot6l*-enhancer region and found that the forced expression of GATA3 increased the *Cnot6l*-enhancer activity (Supplementary Fig. 12f). Intriguingly, the expression of *Cnot6*, a paralog of *Cnot6l*, was decreased in ILC2s in WT and G3SEKO mice (Supplementary Fig. 12g), whereas the expression of *Cnot6l* was increased in WT mice but not G3SEKO mice (Supplementary Fig. 12c), suggesting that the sift from *Cnot6* to *Cnot6l* during ILC2 development is impaired in G3SEKO mice. These results suggest that high levels of GATA3 induced by G3SE are required for ILC2 development, partly through the induction of *Cnot6l* expression.

### G3SEKO mice lack functional ILC2s under allergic inflammatory conditions

Under inflammatory conditions, ILC2s not only migrate from the BM and other organs but also develop from ILCPs in inflamed tissues[2,18,20,43]. To assess the roles of G3SE in ILC2 development under inflammatory conditions, we first utilized an IL-33-induced allergic airway inflammation model (Fig. 6a). Eosinophil numbers in the BALF and lung were significantly decreased in G3SEKO mice compared to those in WT mice (Fig. 6b, c), consistent with a previous report of GATA3+672/762ᐃ/ᐃ mice, which lack a region nearly identical to the SE1 region[27]. ST2⁺ ILCs, KLRG1⁺ ILCs, and IL-5- or IL-13-producing ILCs were also significantly reduced in G3SEKO mice (Fig. 6d).

To further examine the role of G3SE in vivo, we also employed a papain-induced allergic airway inflammation model (Fig. 6e). Consistent with the results of the IL-33-induced model, eosinophils were significantly reduced in the BALF and lung (Fig. 6f, g), and ILCs lacked ILC2 marker expression and produced a minimal amount of type 2 cytokines in G3SEKO mice (Fig. 6h). These data imply that G3SE is required for peripheral ILC2 development under inflammatory conditions as well as in steady-state conditions.

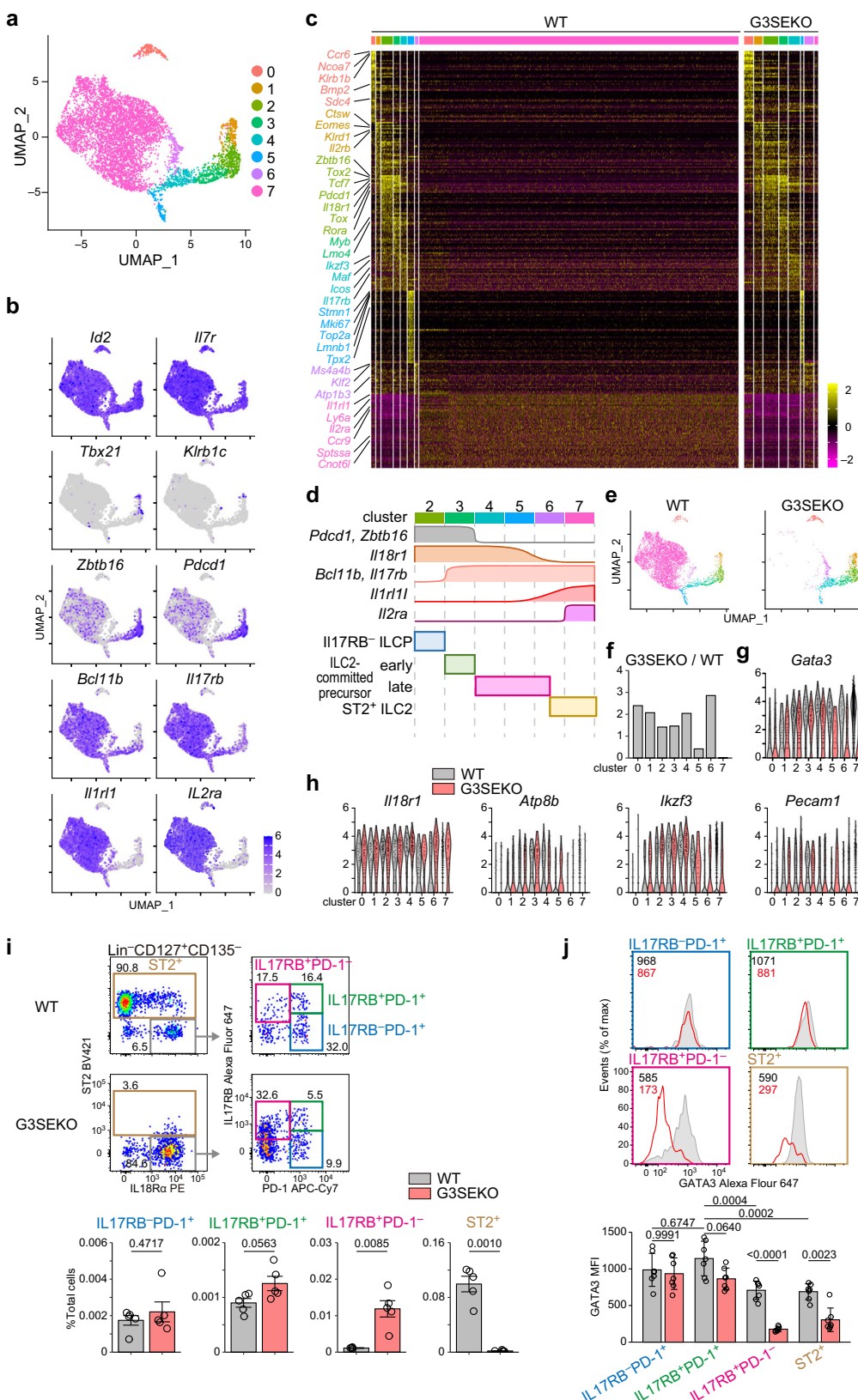

## Mice lacking the SE2 region show reduced ILC2s in peripheral tissues but not in BM

It has recently been reported that GATA3+672/762$^{\Delta/\Delta}$ mice exhibit a 75% reduction of lung ILC2s, 68% of small intestinal ILC2s, and 89 % of BM ILC2s[27], suggesting that the SE1 region has more significant impact on ILC2s in BM than peripheral tissues. In contrast, G3SEKO mice

showed almost complete deficiency of ILC2s in the BM and peripheral tissues. To further our understanding of the roles of SE1 and SE2 regions during ILC2 development, we generated mice lacking the SE2 region (G3SE2KO mice, Fig. 7a). G3SE2KO mice showed reduced numbers of lung and liver ILC2s by 67% and 58%, respectively (Fig. 7b, c). In contrast, G3SE2KO mice showed normal ST2$^+$ ILC2

**Fig. 3 | G3SE is required for ILC2 development following ILC2-lineage commitment. a–h,** Lin⁻CD127⁺CD135⁻LPAM-1⁺ cells from the BM of WT mice and G3SEKO mice were sorted, stained with antibodies and a hashtag (MHC-I), and analyzed by single-cell RNA sequencing (scRNA-seq). **a** Uniform manifold approximation and projection (UMAP) showing 8 clusters. **b** Feature plots showing the expression of ILC development-related genes. **c** Heatmap showing the z-scored expression of the top 50 differentially expressed genes among clusters. **d** Scheme of relationship among the cluster, the expression of marker genes, and the corresponding population during ILC2 development from ILCPs. **e** UMAP showing the cell distribution of WT mice and G3SEKO mice. **f** Ratio of G3SEKO cells/WT cells in each cluster. **g** Violin plot showing *Gata3* expression in the indicated clusters in

each genotype. **h** Violin plots showing marker genes of cluster 3 that were highly expressed in cluster 5 in G3SEKO cells. **i, j** Lin⁻CD127⁺CD135⁻ BM cells were analyzed by flow cytometry. Lin: CD3ε, CD4, CD5, CD8a, CD11b, CD11c, CD19, CD135, NK1.1, Gr1, B220, Ter119. **i** Flow cytometry analysis showing the proportion of the indicated fractions; n = 5. **j** GATA3 expression in cells of the indicated fractions, n = 7. Gray/black area: WT cells; red area: G3SEKO cells; black number: mean fluorescence intensity (MFI) of WT cells; red number: MFI of G3SEKO cells. **i-j** Each data point indicates one mouse from five (**i**) or four (**j**) independent experiments. Data are presented as mean ± SE. Statistical analysis was performed using unpaired, two-sided Welch's *t*-test (**i**) and two-way ANOVA with Tukey's multiple comparisons test (**j**). p values are shown on the graphs. Source data are provided as a Source Data file.

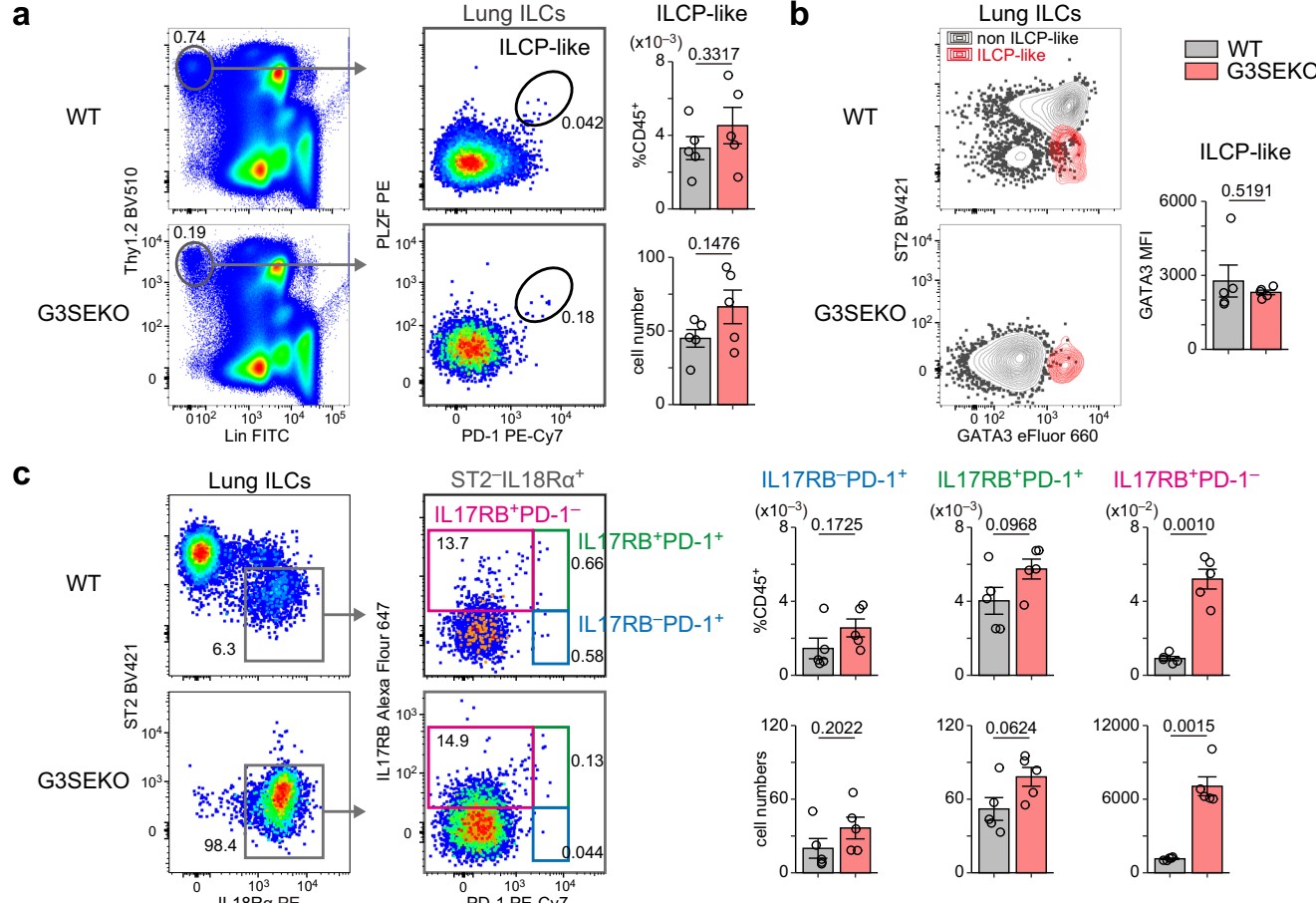

**Fig. 4 | G3SE is not required for high levels of GATA3 in lung ILCP-like cells. a** Flow cytometry analysis of lung ILCP-like cells (CD45⁺Lin⁻Thy1.2⁺PD-1⁺PLZF⁺ cells) in WT and G3SEKO mice, n = 5. Lin: CD3ε, CD4, CD5, CD8a, CD11b, CD11c, CD19, CD49b, Gr1, B220, Ter119. **b** ST2 and GATA3 expression in ILCP-like cells, n = 5. **c** Flow cytometry analysis showing the proportion of the cells in the indicated

fractions, n = 5. **a–c** Each data point represents one mouse from three independent experiments. Data are presented as mean ± SE. Statistical analysis was performed using unpaired, two-sided Welch's *t*-test. p values are shown on the graphs. Source data are provided as a Source Data file.

development in the BM (Fig. 7d). The development of liver ILC1s and ILC3s was not affected by the absence of the SE2 region (Fig. 7c). Consistently, GATA3 expression in the lung and liver ILC2s was modestly decreased, while GATA3 expression in ST2⁺ ILCs in the BM was almost normal in G3SE2KO mice (Fig. 7e–g). These results indicate that SE2 is pivotal in ILC2 development and GATA3 expression in peripheral tissues.

## Discussion

This study revealed that high-GATA3 expression essential for ILC2 development was achieved in the late ILC2-committed precursor stage

via G3SE. Loss of G3SE resulted in the complete absence of mature ILC2s in the lung, liver, and small intestine, in parallel with the accumulation of late ILC2-committed precursors in the BM and lung. High levels of GATA3, induced by G3SE, contributed to ILC2 development partly through *Cnot6l* induction.

We demonstrated the sequential development of ILCPs, early and late ILC2-committed precursors, and ILC2 using scRNA-seq. ILC2-committed precursors contained two stages: early ILC2-committed precursors expressing *Zbtb16* and *Pdcd1* and late ILC2-committed precursors not expressing *Zbtb16* and *Pdcd1*. Xu et al. reported that both populations mainly give rise to ILC2s but not other ILC lineages[12].

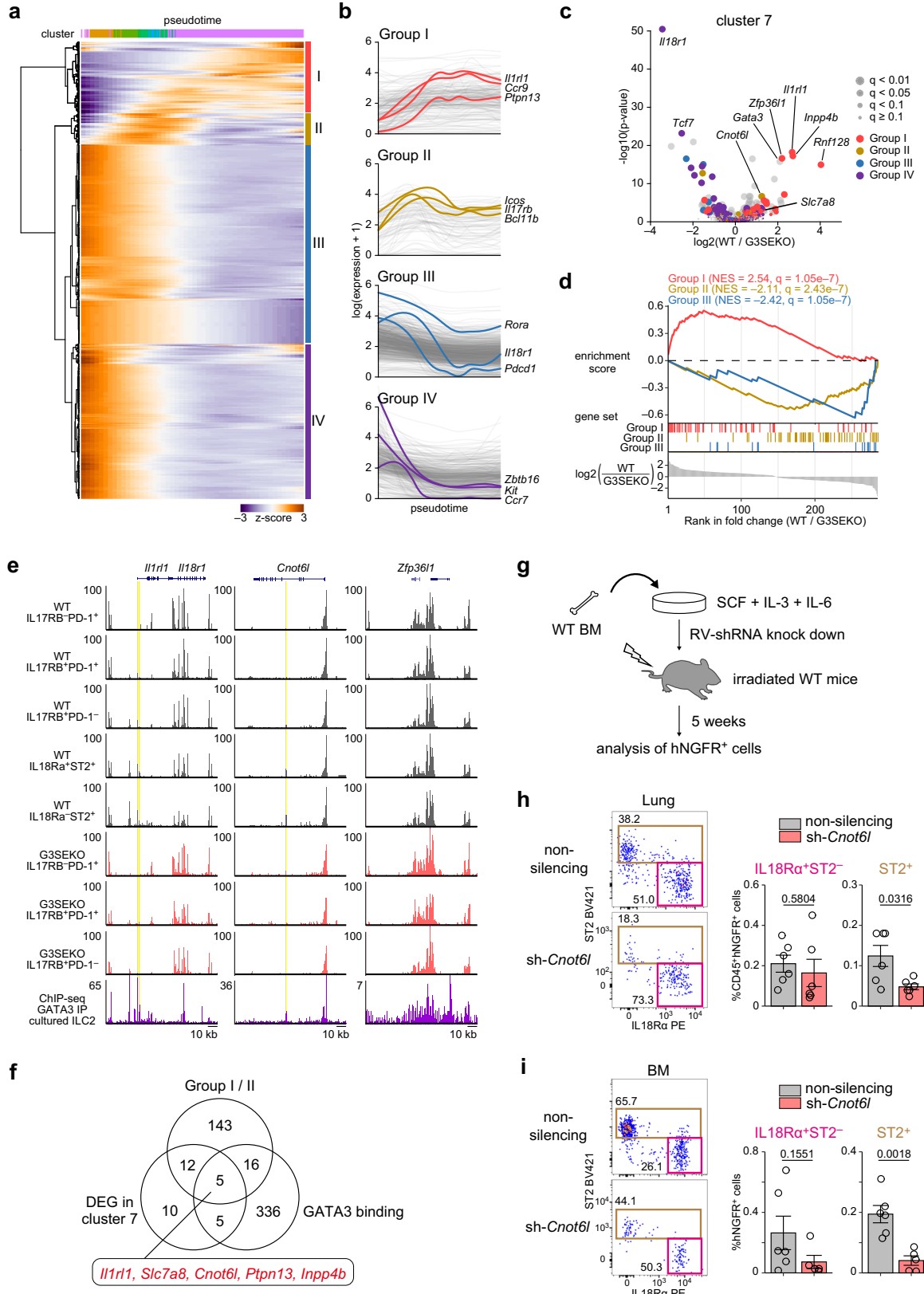

We found that early ILC2-committed precursors express high levels of GATA3 mainly through G3SE-independent mechanisms. With development toward late ILC2-committed precursors, high levels of GATA3 begin to depend on G3SE, resulting in the acquisition of ILC2 features and the loss of ILCP features. These results indicate that early ILC2-committed precursors and late ILC2-committed precursors exhibit

distinct transcriptional regulations, especially regarding GATA3 expression. It is difficult to accurately distinguish whether the decrease in ILC2s in G3SEKO mice was due to a failed transition to ILC2s or a survival issue after becoming ILC2s, as the inducible deletion of GATA3 in ILC2s significantly decreases ILC2 numbers in vitro and in vivo[15,26]. The increase in late ILC2-committed precursors in the BM and the lung

**Fig. 5 | *Cnot6l* is a possible target of GATA3 during ILC2 development.**
**a** Expression profiles of ILC2 development-related genes were calculated using tradeSeq based on pseudotime analysis (See also Supplementary Fig. 12a). Heatmap showing z-scored gene-expression kinetics. Genes were clustered into four groups using hierarchical clustering. **b** Gene expression in each group. Colored lines show indicated genes. **c** Differentially expressed genes (DEGs) between WT and G3SEKO cells in cluster 7. Colors indicate the groups in which the genes were clustered. q < 0.05 & fold change >2, n = 68. **d** Gene set enrichment analysis of group I, II, and III genes in the DEGs in cluster 7. NES: normalized enrichment score; q-value: false discovery rate. **e** UCSC genome browser tracks for ATAC-seq of BM ILC2 lineages and ChIP-seq of GATA3 in cultured ILC2s (GSE111871). Yellow boxes: overlapping regions of newly opened chromatin from ILCPs to ST2⁺ ILC2s in WT mice but not in

G3SEKO mice, accompanied by GATA3 binding. **f** Venn diagram of the number of Group I/II genes, DEGs in cluster 7, and genes with GATA3 binding to open chromatin regions in ILC2s. **g** Experimental procedure for shRNA-transduced BM chimeric mice. Non-silencing or sh-*Cnot6l* retrovirus-transduced WT BM cells were transferred into irradiated recipient mice. Five weeks later, lung cells and BM cells were harvested, and shRNA-transduced cells were analyzed by gating hNGFR⁺ cells. **h** Lung ILC frequency (CD45⁺hNGFR⁺Lin⁻Thy1.2⁺ cells) in CD45⁺hNGFR⁺ cells. n = 6. **i** BM ILC frequency (hNGFR⁺Lin⁻CD127⁺CD135⁻ cells) in hNGFR⁺ cells. non-silencing: n = 6, sh-*Cnot6l*: n = 5. (**h**, **i**) Each data point indicates one mouse from four independent experiments. Data are presented as mean ± SE. Statistical analysis was performed using unpaired, two-sided Welch's *t*-test. p values are shown on the graphs. Source data are provided with this paper.

of G3SEKO mice indicates a failed transition toward ILC2s rather than a maintenance defect.

GATA3 is critical in early ILC development, especially in BM ILCPs[13,22,26,33]. Our data demonstrate that BM ILCPs and lung ILCP-like cells express high levels of GATA3 in G3SEKO mice, whereas late ILC2-committed precursors had reduced GATA3 expression. Therefore, the induction and maintenance of GATA3 in BM ILCP and lung ILCP-like cells is induced via a mechanism that G3SE does not mediate. This induction is stage-specific and lost in the late ILC2-committed precursor stage, where G3SE plays a vital role in GATA3 induction. Consequently, G3SE is required to enter the ILC2 stage.

The number of ILC1s in peripheral tissues was increased in G3SEKO mice, although the number of ILC3s was similar to that in WT mice. It is well known that ILC2s can be converted to ILC1s through stimulation with a combination of IL-12 and IL-1β or IL-18[44,45]. Since G3SE is not active during the ILCP stage in the BM, the increase in ILC1s in G3SEKO mice is likely derived from late ILC2-committed precursors rather than directly from ILCPs. Further study, for example, through the utilization of lineage-tracing of the ILC2-committed precursors in peripheral tissue, is required to clarify the mechanisms underlying ILC1 increases in G3SEKO mice.

The impacts of the *Gata3* enhancer regions on immune systems have been intensively investigated[24,27]. For instance, the enhancer region located 278–285 kb downstream from *Gata3* is reported to play crucial roles in CD4 T cell development[24] and ILC1 and ILC2 development[27]. However, the lack of a 278–285 kb region did not affect the expression of GATA3 and functions of mature ILC2s[27]. On the other hand, *Gata3*+674/762[Δ/Δ] mice, which lack almost identical region to the SE1 region (678–784 kb), showed reduced ILC2 numbers and reduced GATA3 levels in ILC2s in both BM and peripheral tissues. *Gata3*+674/762[Δ/Δ] mice also showed impaired ILC2-dependent allergic inflammation[27]. However, the reduction of ILC2s in *Gata3*+674/762[Δ/Δ] mice is incomplete. In this study, we showed that G3SEKO mice exhibited a complete absence of ILC2s in the BM and peripheral tissues. We also demonstrated that G3SE2KO mice showed reduced ILC2 numbers and reduced GATA3 levels in ILC2s in the peripheral tissues but maintained intact ILC2 development in the BM. These findings suggest that SE1 and SE2 act in concert to develop ILC2 in peripheral tissues and that SE1 but not SE2 plays pivotal roles in ILC2 development in the BM. This notion is consistent with a recent study employing high-resolution chromosomal conformation capture, showing that the regions identical to SE1 and SE2 bound to the *Gata3* TSS in ILC2s but not in ILC1s or ILC3s[46].

Regarding the SE1 region, Kasal et al. subdivided *Gata3*+674/762 region into *Gata3*+674/710 and +710/762 regions and generated mice deficient in each region. They found that *Gata3*+710/762[Δ/Δ] mice showed only a mild reduction of ILC2 number in the lung, and *Gata3*+674/710[Δ/Δ] mice showed normal ILC2 number in the lung, indicating that *Gata3*+674/710 and +710/762 regions redundantly induce the development of ILC2s in the lung[27]. Together with our findings, G3SE contains multiple enhancer regions contributing to

ILC2 development, and the deletion of the entire G3SE region results in the complete absence of ILC2s in the BM and peripheral tissues.

We identified *Cnot6l* as one of the key targets for GATA3-induced ILC2 development. CNOT6L is a subunit of the CCR4-NOT deadenylase complex and is known to promote mRNA decay. We found that the expression of *Cnot6l* was increased in clusters 5-7 in WT cells but was reduced in G3SEKO cells (Supplementary Fig. 12c). The enhancer region in *Cnot6l* locus became accessible in the late ILC2-committed precursors in WT mice but remained closed in G3SEKO mice (Fig. 5e). Furthermore, GATA3 overexpression increased the enhancer activities (Supplementary Fig. 12f). These results indicate that GATA3 induction by G3SE results in the elevation of *Cnot6l* in the late ILC2-committed precursors. Notably, *Cnot6*, a paralog of *Cnot6l*, was decreased in ILC2s in WT mice (Supplementary Fig. 12g), indicating that the CNOT6/CNOT6L expression skews toward CNOT6L in ILC2s. Since the elimination of ILCP-related genes was delayed in G3SEKO ILC2s, it is possible that CNOT6L decays ILCP-related genes and accelerates lineage determination. To understand the precise function of CNOT6L in ILC2 development, further detailed analyses of mice lacking GATA3-binding sites in the *Cnot6l* region or IL17RB⁺ cell-specific conditional *Cnot6l* knockout mice are necessary.

We have identified five candidate genes targeted by GATA3 during ILC2 development. Two of these genes are already known to play pivotal roles in ILC2 development, and we proposed a role for Cnot6l in this process. The remaining two genes did not significantly affect ILC2 development in our experimental setting; however, given the incomplete suppression in the sh-knockdown experiment, further validation of these genes, including Cnot6l, using more robust methodologies, such as genetically deficient mice, is warranted.

In conclusion, our findings demonstrate how ILC2 differentiation is influenced by GATA3 produced following ILC2-lineage commitment. ILC2s have a pathogenic role in severe allergic diseases and organ fibrosis, but there is currently no treatment specific to ILC2 regulation. Therefore, additional research into the ILC2-specific high-GATA3 regulation demonstrated in this study might lead to the development of novel therapeutic options in the future.

## Methods

### Ethics statement

All animal procedures used in this study were approved by the Chiba University Animal Care and Use Committee.

### Mice

C57BL/6J mice were purchased from CLEA Japan Inc. (Tokyo, Japan), and CD45.1 mice were purchased from RIKEN BRC (Ibaraki, Japan). G3SEKO mice and G3SE2KO mice were generated as described below. Mice were housed in micro-isolator cages under specific pathogen-free conditions. The cages were kept at a temperature of 23 ± 3 °C, a humidity of 50 ± 10%, and a 12 h/12 h dark light cycle. Six- to twelve-week-old male and female mice were used in this study. Female mice

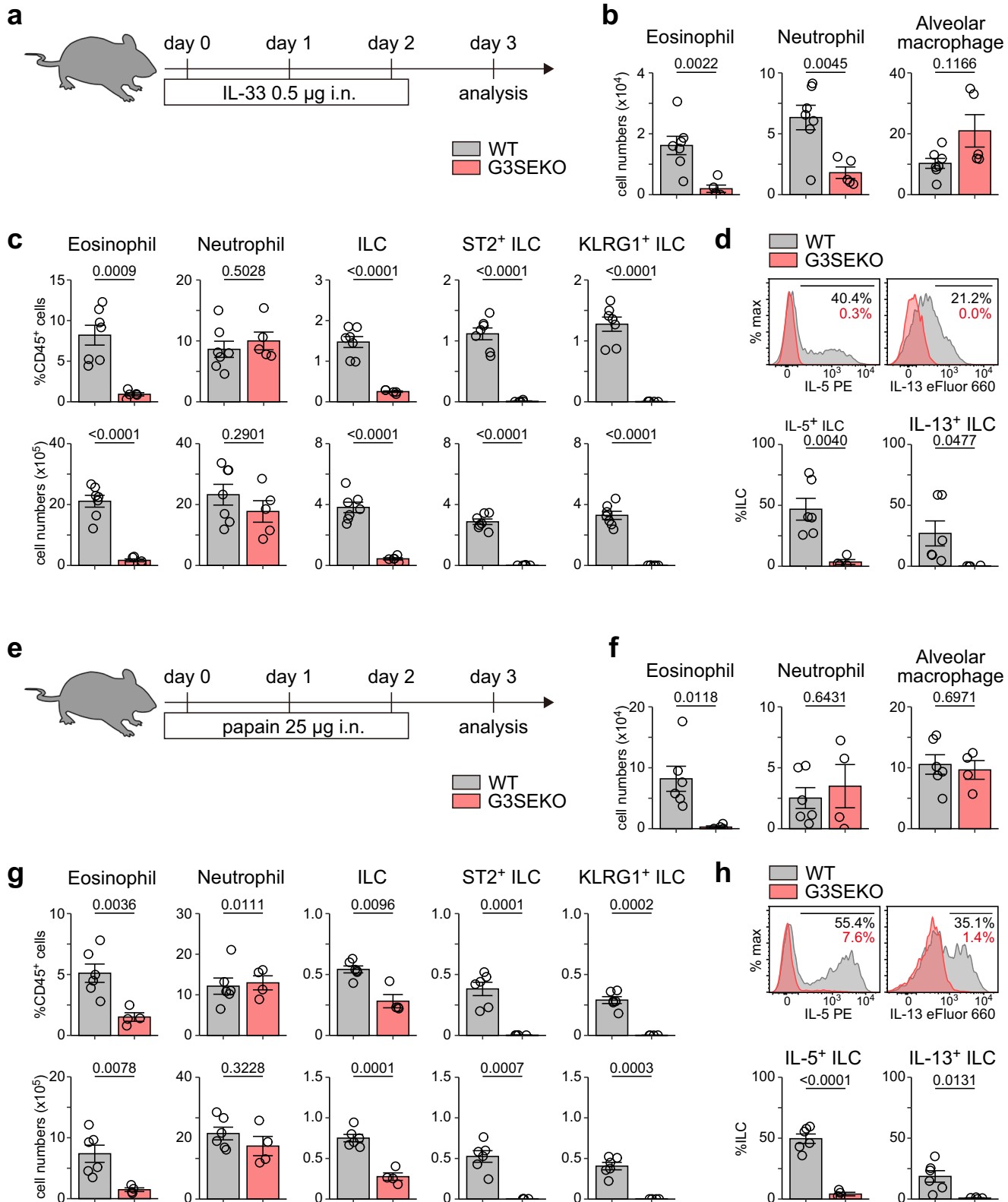

**Fig. 6 | G3SEKO mice lack functional ILC2s under allergic inflammatory conditions. a** Experimental procedure of IL-33-induced allergic inflammation. i.n. intranasal. **b** Eosinophil, neutrophil, and alveolar macrophage count in bronchoalveolar lavage fluid (BALF) of WT and G3SEKO mice with IL-33-induced allergic inflammation. WT: n = 7, G3SEKO: n = 5. **c** Frequencies of indicated cell populations among lung CD45+ cells and their numbers. WT: n = 7, G3SEKO: n = 5. **d** Lung cells isolated from mice subjected to IL-33-induced allergic inflammation were stimulated with phorbol 12-myristate 13-acetate and ionomycin for 5 h. IL-5 and IL-13

production by ILCs (CD45+Lin−Thy1.2+ cells) was evaluated by flow cytometry. WT: n = 6, G3SEKO: n = 5. **e–h** Scheme (**e**), cell counts in BALF (**f**), and frequencies of indicated cell populations among lung cells and their numbers (**g, h**) of WT and G3SEKO mice with papain-induced allergic inflammation. WT: n = 6, G3SEKO: n = 4. (**b–d, f–h**) Each data point indicates one mouse from four (**b–d**) or two (**f–h**) independent experiments. Data are presented as mean ± SE. Statistical analysis was performed using unpaired, two-sided Welch's t-test. p values are shown on the graphs. Source data are provided as a Source Data file.

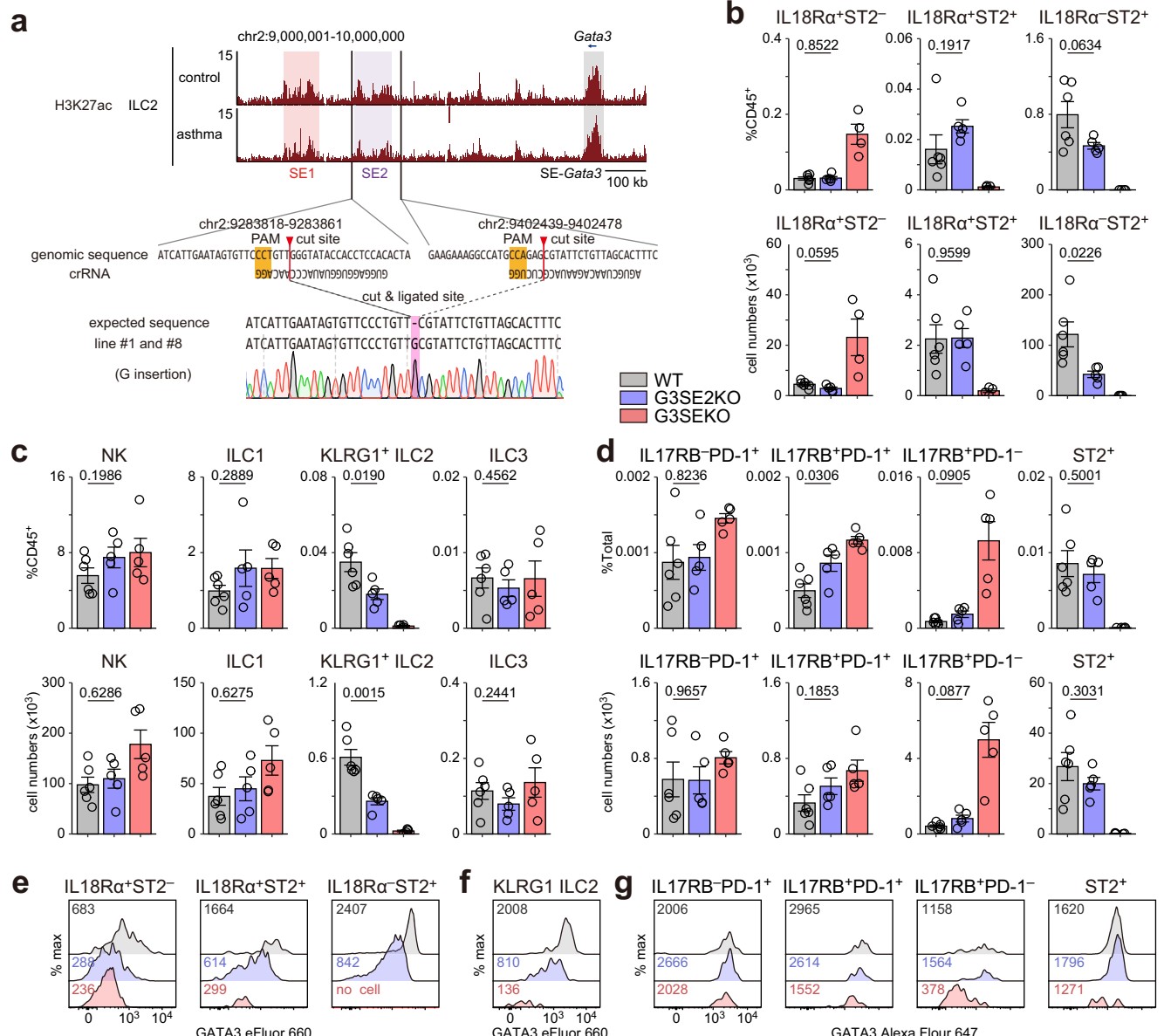

**Fig. 7 | G3SE2KO mice showed reduced peripheral ILC2s and GATA3 expression.**
**a** Upper panel: UCSC genome tracks around the Gata3 and G3SE region. Middle panel: Genomic sequence of upstream/downstream crRNA target regions. Lower panel: DNA sequences of G3SE2 deleted loci from two lines of generated G3SE2KO mice (both G insertion). **b**–**d** Lung (**b**), liver (**c**), and BM (**d**) ILC lineages. Each data point indicates one mouse from two independent experiments. WT: n = 6,

G3SE2KO: n = 5, Lung G3SEKO: n = 4, Liver and BM G3SEKO: n = 5. Data are presented as mean ± SE. Statistical analysis was performed between WT and G3SE2KO mice using unpaired, two-sided Welch's t-test. p values are shown on the graphs. **e**–**g**, GATA3 expression of indicated cells in lung (**e**), liver (**f**), and BM (**g**). Data were representative of two independent experiments. Numbers indicated the mean fluorescence intensity of GATA3. Source data are provided as a Source Data file.

were used for allergic airway inflammation models, as it is known that phenotype is attenuated in male mice.

### Generation of G3SEKO mice and G3SE2KO mice

To form ribonucleoprotein complexes (RNPs), the gRNAs flanking the G3SE region (chr2; 9101999-9102118 and chr2; 9402399-9402518, Supplementary Table 1 and Supplementary Fig. 2) and G3SE2 region (chr2:9283835-9283857 and chr2:9402453-9402475, Supplementary Table 1 and Fig. 7a) were annealed with tracrRNA (IDT, Cat#1072533) and bound to Cas9 protein (Thermo Fisher Scientific, Cat#A36498). Oocytes isolated from C57BL/6J female mice pretreated with pregnant mare's serum (ASKA Pharmaceutical, SEROTROPIN) (7.5 units/i.p.) and human chorionic gonadotropin (ASKA Pharmaceutical, GONA-TROPIN3000) (7.5 units/i.p.) were in vitro fertilized with C57BL/6J mice

sperms. Zygotes were washed twice with Opti-MEM and placed in the electrode gap of the electrode chamber filled with 5 μl of RNP. Electroporation was carried out using Genome Editor TM (BEX) on the following conditions: 25 V, 3 ms ON +97 ms OFF ×8 times. After electroporation, eggs were cultured for 24 h, and two-cell stage embryos were transferred to the oviduct of pseudopregnant ICR females. The genotype of delivered mice was assessed with TOPO TA cloning (Thermo Fisher Scientific, Cat#45-0641) and Sanger sequences (Supplementary Fig. 2 and Fig. 7a). To analyze off-target editing, we first perform CAS-OFFinder[47] with the parameters "mismatch number 2, DNA bulge size 1, RNA bulge size 1". We found 26 candidates for downstream crRNA and 7 candidates for upstream crRNA in G3SEKO mice. T7 analyses were performed for mixed equimolar DNA of G3SEKO and C57BL/6 WT mice (Supplementary Fig. 3, Supplementary Table 1).

## Cell isolations

Single-cell suspensions of the lungs were generated as previously described[48]. Briefly, the lungs were cut into small pieces and digested with Collagenase A (1 mg/ml, Merck, Cat#A36498) and DNase I (0.1 mg/ml, Wako, Cat#043-26773) using gentleMACS Dissociator (Miltenyi Biotec). BM cells were harvested by centrifugation of femurs, tibias, pelvis, humerus, and scapula. Splenocytes and thymocytes were isolated by physical dissociation. The liver was minced and digested by Liberase TM (0.05 mg/ml, Roche, Cat#5401127001) and DNase I (0.2 mg/ml) using gentleMACS Dissociator after 10 ml PBS perfusion. Hepatic leukocytes were isolated from the interface between 40% and 80% Percoll gradient. The small intestine was washed, opened longitudinally, and cut into pieces. The samples were incubated with HEPES (15 mM, Thermo Fisher Scientific, Cat#15630080), EDTA (5 mM, Thermo Fisher Scientific, Cat#AM9260G), DTT (1 mM, Wako, Cat#048-29224), then washed, minced, and digested by Liberase TL (0.25 mg/ml, Merck, Cat#5401020001) and DNase I (0.3 mg/ml). Intestine leukocytes were purified using the Percoll gradient as above. Single-cell suspensions were passed through a 30 μm filter before downstream experiments.

## Allergic airway inflammation

Female mice were anesthetized with an intraperitoneal injection of midazolam, butorphanol, and medetomidine before treatment. For HDM-induced allergic airway inflammation, mice were intratracheally administered 10 μg of HDM (Greer Laboratories, Cat#NC1468783) in 25 μl of PBS. Seven days later, mice were challenged with HDM (10 μg) for five consecutive days. Forty-eight hours after the last challenge, BALF and lungs were harvested. For ovalbumin (OVA, Merck, Cat#A5503)-induced allergic airway inflammation, mice were immunized intraperitoneally (i.p.) with 50 μg OVA absorbed in aluminum hydroxide gel (Thermo Fisher Scientific, Cat#77161) on day 0, 7, and 14 and intranasally challenged with 100 μg OVA on day 20, 21, and 22. Twenty-four hours after the last challenge, BALF and lungs were harvested. For IL-33- or papain-induced allergic airway inflammation, mice were intranasally administered 500 ng of murine IL-33 (BioLegend, Cat#580506) or 25 μg of papain (Nacalai Tesque, Cat#260356-92) in 25 μl of PBS for three consecutive days. Twenty-four hours after the last challenge, BALF and lungs were harvested.

## Mixed BM chimeras

BM cells ($2.5 \times 10^6$) from CD45.2$^+$ G3SEKO mice and CD45.1$^+$ WT mice were mixed and injected intravenously into CD45.1$^+$CD45.2$^+$ WT mice 4 h after total body irradiation (9.5 Gy). Ten to twelve weeks later, mice were analyzed.

## Flow cytometry and cell sorting

Information on antibodies is shown in Supplementary Table 2. Cells were stained with a Zombie dye (BioLegend, Cat#423106 and 423112), and the indicated antibodies and then analyzed using FACSCanto II (BD Biosciences) or LSRFortessa (BD Biosciences). For transcription factor-staining, cells were fixed and permeabilized with a Foxp3 Staining Buffer Set (Thermo Fisher Scientific, Cat#00-5523-00). For intracellular cytokine-staining, cells were stimulated with Phorbol 12-myristate 13-acetate (Merck, Cat#524400) and ionomycin (Merck, Cat#407950) in the presence of monensin (Merck, Cat#475895) for 5 h, followed by fixation and permeabilization with a Fixation/Permeabilization Solution Kit (BD Biosciences). In some experiments, cells were isolated with SH800 cell sorter (Sony Biotechnology), FACSAria II (BD Biosciences), or FACSMelody (BD Biosciences).

## T cell culture

Naïve CD4 T cells were harvested from lymph nodes and spleens using a CD4$^+$CD62L$^+$ T cell isolation kit II (Miltenyi Biotec, Cat#130-106-643) and stimulated with plate-bound anti-CD3ε Ab (1 μg/ml, BioXCell, Cat#BP0001-1) and soluble anti-CD28 Ab (2 μg/ml, BioXCell, Cat#BP0015-1) under Th0 (no cytokine nor antibody), Th1 [IL-12 (10 ng/ml, Wako, Cat#095-05331) and anti-IL-4 (10 μg/ml, BioXCell, Cat#BP0045)], and Th2 [IL-4 (10 ng/ml, Peprotech, Cat#214-14) and anti-IFN-γ (10 μg/ml, BioXCell, Cat#BP0055)] conditions for 3 days.

## In vitro ILCP culture on OP9-DL1 stromal cells

ILCPs were cultured as previously described[49]. Briefly, OP9-DL1 cells were cultured in 48-well plates at a density of $3 \times 10^4$ cells per well. Twenty hours later, OP9-DL1 cells were treated with Mitomycin C (50 μg/ml, Merck, Cat#M4287) for 25 min. ILCPs were sorted from BM after enrichment with a Mouse Hematopoietic Progenitor Cell Isolation kit (BioLegend, Cat#480004) and seeded onto OP9-DL1 monolayers with ILC medium [DMEM supplemented with 10% FCS, 1% penicillin, 1% streptomycin, 20 μM HEPES, 1 μM sodium pyruvate, 1× non-essential amino acids, 20 ng/ml IL-7 (BioLegend, Cat#577802) and 50 ng/ml SCF (BioLegend, Cat#579704)]. An equal volume of fresh ILC medium was supplemented three days later. Five days after the initial culture, cells were harvested and analyzed.

## Retrovirus-mediated gene expression

MSCV-GATA3-IRES-Thy1.1 was kindly provided by Dr. Kenneth Murphy (Washington University, St. Louis). To express genes in cultured ILCP using a retrovirus, cells were cultured for 24 h, transferred to a RetroNectin-coated plate with viruses, centrifuged at $500 \times g$ for 90 min at 32 °C, and then cultured on OP9-DL1 monolayers for another four days.

## Retrovirus-mediated gene knockdown

MSCV-LTR-miR30-puro-IRES-GFP (LMP) was obtained from Thermo Fisher Scientific (Cat#EAV4678). The GFP cassette of the LMP vector was substituted with the human NGFR. The shRNA sequences targeting *Cnot6l*, *Ptpn13*, and *Inpp4b* were designed using GPP Web Portal (https://portals.broadinstitute.org/gpp/public/gene/search), siDirect version 2.0 (http://sidirect2.rnai.jp/), and three shRNA sequences for each target were manually curated. Knockdown retrovirus vectors were made by subcloning the shRNA sequence into the LMP vector according to the manufacturer's instructions. Knockdown efficiency was validated using in vitro T cell culture systems, and the most efficient shRNA sequences for each gene are shown in Supplementary Table 1. Retrovirus-mediated gene knockdown for BM chimeric experiments was performed as described previously[50]. Briefly, BM cells were cultured overnight in IMDM supplemented with 15% FBS, IL-3 (10 ng/ml, Peprotech, Cat#213-13), IL-6 (5 ng/ml, Miltenyi Biotec, Cat#130-096-684), and SCF (100 ng/mL). Cells were then spin-infected with retroviral supernatants supplemented with Polybrene (10 μg/ml, Nacalai Tesque, Cat#12996-81). Twenty-four hours later, cells were washed and transferred to irradiated C57BL/6J recipients. Mice were analyzed five weeks after BM transfer.

## Luciferase reporter assay

Luciferase reporter assay was performed as previously described with minor modifications[51]. *Cnot6l*-enhancer region (chr5:96111540-96112644) was cloned into pGL4.23[luc2 minP] vector (Promega, Cat#E8411). 293T cells were transfected with either pGL4.23-empty-enhancer vectors or pGL4.23-*Cnot6l*-enhancer vectors and either MSCV-empty-ires-Thy1.1 vectors or MSCV-*Gata3*-ires-Thy1.1 vectors in the presence of pGL4.74[hRluc/TK] vectors (Promega, Cat#E6921). Twenty-four hours later, RLUs were assessed with a dual luciferase assay system (Promega, Cat#E1910). The enhancer activity was normalized by the mean of RLU signals of MSCV-empty-IRES-Thy1.1 vector-transduced conditions.

## ChIP-seq analyses

ChIP-seq analyses were performed as previously described[52]. Briefly, cells were stained and crosslinked with 1% formaldehyde at room temperature for 5 min before FACS sorting. Sorted CD4 T cells (CD4$^+$CD8$^-$CD11c$^-$CD49b$^-$TCRγδ$^-$EpCAM$^-$) and ILC2 cells (Lin$^-$Thy1.2$^+$ST2$^+$, Lin: CD4, CD8a, CD19, FcεRI, TCRβ, Gr1, CD11b, CD5 and Ter119) were sonicated with Picoruptor (Diagenode) and were immunoprecipitated by Dynabeads Protein A (Thermo Fisher Sciences, Cat#10001D) preincubated with anti-H3K27ac antibody (Abcam, Cat#ab4729). Beads were washed seven times and reverse-crosslinked by overnight incubation at 65 °C. DNA fragments were harvested with phenol-chloroform extraction followed by ethanol precipitation. ChIP-seq libraries were prepared with a ThruPLEX DNA-seq kit (Takara Bio, Cat#R400675) and were sequenced with an Illumina HiSeq 2500 in a 50 bp single-end mode. ChIP-seq data were aligned to mm10 genome assembly using Bowtie2[53] and were masked with a blacklist of ENCODE project[54]. Data were visualized using the HOMER package[55] and UCSC genome browser. SEs were identified from H3K27ac ChIP-seq data using findPeaks in the HOMER package with the following parameter, -style superhistone, -poisson 1e-8, and -excludePeaks option for excluding TSS ± 2500 kb region. Tag counts on merged SEs were generated using HOMER. Normalized tag counts on SEs were calculated by subtracting input ChIP-seq data and TMM normalization. ChIP-seq data for GATA3 in cultured ILC2 with IL-2 and IL-7 (GEO: GSE111871)[37] and ChIP-seq data for Bcl11b in ILC2/b6 cell line (GEO: GSE131082)[56] were re-analyzed from raw data. GATA3-binding peaks were identified using findPeaks with -style factor -poisson 1e−8 -size 300.

## ATAC-seq analysis

ATAC-seq of BM ILCs was performed using the Omni-ATAC protocol, as previously described[57]. Briefly, cells were incubated and washed with ATAC-Resuspension Buffer. Transposition reactions were performed using Illumina Tagment DNA Enzyme and Buffer Small Kit (Illumina, Cat#20034197), followed by DNA clean-up using DNA Clean and Concentrator-5 kit (Zymo Research, Cat#D4014). Libraries were amplified using a NEBNext High-Fidelity 2× PCR Master Mix (New England Biolabs, Cat#M0541S) with an Ad1 universal primer and Ad2.1-Ad2.5 index primers (Supplementary Table 1). Sequencing was performed with an Illumina NextSeq 500 in 75 bp paired-end mode. ATAC-seq data of BM ILCs were aligned with paired-end mode, and the index of the paired-end was trimmed using the SeqPurge program. Published ATAC-seq data of cultured helper T cell subsets were from GEO: GSE159505[31]. ATAC-seq data of cultured T cells were aligned to mm10 genome assembly using Bowtie2 with single-end mode. Mapped fragments were trimmed 4 bp from the start and 5 bp from the end using bedtools[58]. Peaks were identified using findPeaks with -style factor, -poisson 1e−8, and -size 300. We considered a 4-fold change in ATAC-seq data to be significant in identifying chromatin regions that opened differently between two conditions.

## Single-cell RNA sequencing library preparation and sequencing

BM cells were harvested from the femur and tibia of ten WT mice and eight G3SEKO mice in the 1st experiment and the femur, tibia, pelvis, scapula, humerus, and forearm of seven WT mice and nine G3SEKO mice in the 2nd experiment. Lineage-positive cells were depleted using a Hematopoietic Progenitor Cell Isolation Kit (BioLegend). Cells were stained with Lin (CD3e, CD4, CD5, CD8, CD11b, CD11c, CD19, Gr1, NK1.1, Ter119, B220), LPAM-1, FLT3, PD-1, CD127, CD25, and hashtag for MHC class I (BioLegend). Lin$^-$CD127$^+$FLT3$^-$LPAM-1$^+$ αLP were sorted using FACSAria II (BD Biosciences) (Supplementary Fig. 4a). Combined cells were trapped and reverse-transcribed using BD Rhapsody (BD Biosciences) according to the manufacturer's instructions. Single-cell RNA sequencing was performed by ImmunoGeneTeqs (Chiba, Japan) using the platform of TAS-Seq[59]. Briefly, cDNA and hashtag libraries were

generated using a KAPA Hifi ReadyMix (Roche, Cat#KK2601) and a NEBNext Ultra II FS library prep kit for Illumina (New England Biolabs, Cat#E7805) and then quantified using a KAPA Library Quantification Kit (Roche, Cat#KK4824). Sequencing was performed by Illumina Novaseq 6000 sequencer to a depth of approximately 100,000 reads per cell.

## Single-cell RNA sequencing data processing

After adapter removal by Cutadpat-v3.4[60], gene-expression libraries were aligned to the Ensembl GRCm38 release-101 by Bowtie2−2.4.2, and read counts were generated using mawk. The inflection threshold of the knee-plot was identified by the DropletUtils package[61]. Sample origins and doublets were identified based on fold change of the normalized read counts of the hashtags. The count matrices were imported and analyzed using Seurat v.4[62]. Cells with more than 10% of mitochondrial genes and less than 83012 tag counts in the 1st experiment and 6733 in the 2nd experiment (inflection point) were filtered out. Genes started by "Gm" and ribosomal protein were excluded from downstream analyses. The genotype of cells was identified using Hashtags. The count data were normalized using SCTransform[63] and then integrated into one Seurat object. The cells with low *Id2* transcript levels were excluded from downstream analyses. Principal-component analysis (PCA) was performed for dimension reduction. The first 8 principal components were used for clustering and uniform manifold approximation and projection (UMAP). We first classified cells into three clusters (ILC2, LTi cells/ILC3, and others) at a low resolution and then performed a sub-clustering on others. Differentially expressed genes were identified using the Wilcoxon test with the q-value of Storey correction. Pseudotime was analyzed using Slingshot[64], and kinetics of gene expression along with pseudotime was calculated using tradeSeq[65]. Gene set enrichment analyses were performed using clusterProfiler[66] in the R package.

## Statistics

Statistical analysis was performed using R software (version 4.0.2). The statistical analysis of the results is described in each figure legend. Data are summarized as mean ± SE. $p < 0.05$ was considered significant.

## Reporting summary

Further information on research design is available in the Nature Portfolio Reporting Summary linked to this article.

## Data availability

All sequencing data generated in this study have been deposited in the GEO database under accession code GSE218144 for ChIP-seq, GSE218142 for ATAC-seq, and GSE218144 for scRNA-seq. Published ATAC-seq data of cultured helper T cell subsets were from GSE159505[31]. Published ChIP-seq data for GATA3 and Bcl11b were from GSE111871[37] and GSE131082[56], respectively. All other data are available in the article and its Supplementary files or from the corresponding author upon request. Source data are provided with this paper.

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

## Acknowledgements

We thank Ms. Kazumi Nemoto, Ms. Maho Yoshino, Ms. Kaoru Sugaya (Chiba University), Dr. Shigeyuki Shichino (Tokyo University of Science), and Dr. Yoshinori Hasegawa (Kazusa DNA Research Institute) for their technical assistance. This work was supported in part by the Japan Agency for Medical Research and Development grant (JP19ek0410043h0003), the Japan Science and Technology Agency (Moonshot R&D) grant (JPMJMS2025), Chiba University Synergy Institute for Futuristic Mucosal Vaccine Research and Development (cSIMVa), Agency for Medical Research and Development (AMED)-SCARDA [223fa627003h0002], and GSK Japan Research Grant 2020. We would like to thank Editage (www.editage.com) for editing the English language.

## Author contributions

All authors were involved in drafting the manuscript and approved the final version for publication. Dr. Iwata has full access to all data in the study and takes responsibility for it. Study conception and design: H.F., Y.T., A.I., and H.N. Performed research and analysis: H.F., Y.T., A.I., M.K., K.K., T.Kumagai., T.Kageyama, S.T., A.Suto, and K.S. Generation of G3SEKO mice and G3SE2KO mice: H.F., A.I., L.F., A.Sakamoto, and M.H. Manuscript preparation: H.F., Y.T., A.I., and H.N.

## Competing interests

The authors declare no competing interests.
