## [Peer Review File · Nature Communications]

Stage-specific GATA3 induction promotes ILC2 development after lineage commitmentREVIEWER COMMENTS

Reviewer #1 T cell transcriptional regulation (Remarks to the Author):

In this study, the authors aimed to understand ILC2 development driven through controlled high expression of GATA3 via cell type specific super enhancers (SE). The authors generated an enhancer deleted mice that were missing 2 tandem GATA3 SE regions (G3SEKO) and evaluated the ILC2 phenotype. G3SEKO cells showed cell intrinsic impairment to become mature ILC2 in tissues, and loss of high GATA3 expression during transition from ILC precursor to mature ILC2 in vitro. In contrast, G3SEs seems dispensable for maintaining high GATA3 expression in ILCP in BM and lung. Single cell RNA-seq analysis of BM cells illustrated possible difference between WT and G2SEKO and guided the authors to identify Cnot6l as a potential downstream target of GATA3 in supporting a later stage of ILC2 maturation. When challenged to induce type 2 inflammation in lung, G3SEKO mice exhibited muted type 2 response due to the lack of ILC2 population in the tissue.

The work clearly demonstrated the importance and the critical time window of the GATA3 super enhancers to support fully developed ILC2 in the tissues. However, the report suffers from the fact that scRNA-seq analyses are based on seriously under-sampled and WT/KO unbalanced data, rendering interpretation of computational analyses unconvincing. Repeating scRNA-seq with more cell capture per genotype and confirming the reproducibility of observations based on WT-KO balanced samples is highly recommended. Also lacking is any validation of intended CRISPR deletions of the target regions, or evaluations of off target editing in their newly generated mouse line. Overall, the report in its current form falls short of being fully developed strong story. However, a relatively straightforward addition of repeat data analyses as well as molecular diagnosis of DNA deletions (SE1 and SE2) will improve the quality of the paper.

Reviewer #2 supervisor for reviewer #3

Reviewer #3 ILC2 (Remarks to the Author):

The authors described a newly identified super enhancer governing GATA3 expression specifically in ILC2s. Deletion of this region in vivo generates a mature ILC2 deficient mouse, and therefore does not mount an allergic response in the lung when exposed to allergens. Their work has revealed a novel layer of regulation for ILC2s which is complementary to other studies showing the requirement of GATA3 for ILC2 development and function. This is an interesting work, well executed and that will contribute to our understanding of the regulation of ILC2s.

I would like to put forward a few comments/questions for the authors:

1. Characterization of the G3SEKO mouse. GATA3 is likely to be expressed in other cells apart from ILCs. What is the distribution of other immune cell types in the tissues analyzed?
2. Figure 1: Quantification of ILCs in the HDM model should be shown.
3. Figure 2. The authors conclude that ILCP CD25- PD1+ cells accumulate in the BM of G3SEKO mice. Could the authors show absolute numbers?
4. Figure 3: Can the authors show gating strategy for the purification of aLPs in the scRNAseq experiment?
5. Figure 3: scRNAseq experiment (Figure 3d-e) does not reflect the accumulation of PD1+ ILC2Ps observed in Figure 2b.
6. Figure 3: Why are there so few G3SEKO aLPs sequenced? Can this alter the DEG analysis downstream?
7. Figure 3: The blockade in ILC2 development is in II18ra+ ST2- ILC2Ps in the G3SEKO mouse. Analysis of the activation of the gata3 SEs is shown in figure 1 for lung ILC2s. Are the Gata3 SEs also active in II18ra+ ST2- ILC2Ps?
8. Figure 4. Can the authors show absolute counts for the data presented in this figure.
9. Figure 4c. Staining for II17rb is dim. The increase in IL17rb+ cells seems to be due to the increase in the double negative population. What are II17rb- Pd1- cells?
10. In Figure 3, ILC2 progenitors are analysed using II17rb and Pd1. In figure 4 in the lung, ILC2 progenitors are gated first using II18r and ST2 markers and then data is shown for IL17rb and Pd1 (Figure 4c). In figure 5, both lung and BM are analyzed in the same manner as IL18ra+ ST2- ILC2P. Keeping the same gating strategy for ILC2P analysis would aid the understanding of the paper.

11. Figure 5: ST2⁺ ILC2s are reduced in shCnotl6 condition. Can the authors show Gata3 expression in shCnotl6 vs shControl, to exclude the possibility of ST2 expression downregulation? Can the authors show absolute numbers?
12. Figure 5: Cnotl6 downregulation does not seem to replicate G3SEKO phenotype at the level of IL18Ra⁺ ST2⁻ progenitors, as it is not increased. Showing absolute numbers may clarify this for the reader.
13. Figure 5: Are there any differences in lung or BM IL17rb⁻ PD1⁺ ILC2s in the absence of Cnotl6?
14. Figure 6: Can the authors show absolute quantification for ILC numbers in both allergy models. Also, can the authors show histograms for IL-5 and IL-13 cytokine staining?

Reviewer #4 ILC2 (Remarks to the Author):

In this study, Furuya, Nakajima, and colleagues characterize a pair of enhancer regions downstream of the Gata3 locus that together drive the elevated Gata3 expression levels necessary for group 2 innate lymphocyte (ILC2) development. The authors first identify the region encompassing super enhancer 1 (SE1) and SE2, which they term GATA3-related tandem super enhancers (G3SE), using ChIP-seq for H3K27 acetylation followed by CRISPR/Cas9-mediated deletion of ~300 kb to generate G3SE knockout (G3SEKO) mice. As expected, in profiling these mice they find that ILC2 populations are dramatically reduced in a cell-intrinsic manner in the lung, liver, and small intestine, and bone marrow (BM) whereas group 1 and group 3 ILCs are unaffected. Furthermore, *in vitro* differentiation of G3SEKO ILC precursors (ILCP) into ILC2s is impaired but is rescued by retroviral expression of Gata3. Next the authors employ scRNA-seq to identify when G3SE is required to induce high levels of Gata3 during ILC2 development in the BM. Consistent with previous reports, they identify four stages of ILC2 development: ILCP, early-ILC2P, late-ILC2P, and ILC2. Comparing WT and G3SEKO scRNA-seq data, the authors find that ILC development in G3SEKO mice is arrested at the late-ILC2P stage, which they further validate by flow cytometry using PD-1, IL-17RB, IL-18R α , and ST2 expression as markers of developmental progression. The authors then examine peripheral ILC2 precursors in the lung and find that, like in the BM, ILCPs arise normally in G3SEKO mice whereas ILC2 development is abolished. Using a screening criterion that includes genes upregulated during ILC2 development, differentially expressed genes between WT and G3SEKO cells, and ATAC-seq of ILCP and ILC2s, the authors identify several genes as GATA3 target candidates during ILC2 development. *In vivo* knockdown of one, Cnot6l, impairs ILC2 development in the lung and BM. Lastly, the authors demonstrate that G3SEKO mice have impaired ILC2 responses during IL-33- and papain-induced allergic airway inflammation.

While many of the findings reported here were previously published in Kasal 2021 PNAS, the methodology and rigor of this manuscript are strong, such that it provides further support for earlier findings. The authors' interpretation and conclusion that G3SE is required to drive Gata3 expression during ILC2 development, specifically at the transition from early- to late-ILC2P, is well supported by their methodological approach and data. Several findings in this manuscript are noteworthy, particularly the proposed involvement of Cnot6l. However, on the whole it currently lacks the novelty necessary to advance the field of ILC2 development. Figures 1 and 6 were fundamentally shown in Kasal 2021 PNAS using Gata3 +672/764 Δ/Δ mice, which is captured in the larger G3SE deletion as SE1. Additionally, much of ILC2 development at single cell resolution shown in Figure 3 has been published before in Yu 2016 Nature, Ishizuka 2016 Nat. Imm., Walker 2019 Immunity, and Xu 2019 Immunity.

Key Results

- G3SE is required to drive high levels of Gata3 expression during the developmental transition from late-ILC2Ps to ILC2s in the BM and lung at homeostasis and during allergic inflammation.
- GATA3 may drive Cnot6l expression during ILC2 development.

Validity

1. Methodological approach, data quality and presentation
 - a. Major

- i. An insufficient number of cells (53) was collected for G3SE knockout α LP scRNA-seq and must be expanded to improve data quality.
- ii. There is insufficient rationale given by the authors as to why SE1, SE2, and G3SE should be described as super-enhancers. If this term is to be used, the authors must demonstrate that the regions are “composed of clusters of active enhancers” and represent a region with characteristics of super enhancers as first described in Whyte 2015 Cell. This includes dense occupation by Mediator, enhancer size, transcription factor density and content, ability to activate transcription, and sensitivity to perturbation.
- b. Minor
 - i. The choice to perform ChIP-seq on total CD4+ T cells sorted from the lung following house dust mite challenge likely dilutes the true Th2 cell signal at SE1 and SE2 from their data in Figure 1A.
- 2. Analytical approach
 - No issues were identified with the statistical analysis.
- 3. Data interpretation and conclusions
 - a. Major
 - i. The claim that G3SE functions partially through Cnot6l is not fully supported. The authors show that Cnot6l expression is decreased in G3SEKO late-ILC2P and that knockdown of Cnot6l also impairs ILC2 development. Nevertheless, without further analysis, such as overexpressing Cnot6l in G3SEKO cells, the authors cannot make the claim.
 - ii. The authors state “Although we did not confirm the respective role of the two tandem SEs, they are more likely to function together ... [Gata3 +672/764 Δ/Δ] was almost identical to our SE1 region (678–784 kb); therefore, the markedly strong reduction in G3SEKO mice is due to the combination of SE1 and SE2 regions. Thus, SE1 and SE2 additively contribute to the GATA3 induction required for ILC2 development.” Without deleting SE2 separately from G3SE, the authors can make no substantive claim as to the function of SE2 and the difference between G3SEKO and Gata3 +672/764 (SE1) KO mice.
 - b. Minor
 - i. The claim of no Th2 cell deficiency is overstated. In vitro differentiation of Th2 cells is insufficient to demonstrate no impact on Th2 cell development particularly given that Kasal 2021 PNAS demonstrated that Gata3 +672/764 (SE1) partially controls Gata3 expression in Th2 cells and is active in Th2 cells.
 - ii. The finding from ATAC-seq data that SE1 is inaccessible in Th2 cells is incongruent with re-analysis of public ATAC-seq data in Kasal 2021 PNAS (Data from Shih 2016 Cell on ILC2s and Th2 cells sorted from active parasitic worm infection). Moreover, the authors own ChIP-seq data for H3K27ac shows binding in SE1 and SE2 in CD4+ T cells sorted after house dust mite exposure but is likely reduced due to diluted Th2 cell signal among total CD4+ T cells.
 - iii. Papain-induced allergic inflammation in the lung is IL-33 dependent (Halim 2014 Immunity). Therefore, the statement that papain would “exclude the possibility that attenuated eosinophilic inflammation was caused by the decreased expression of ST2” is incorrect.

Clarity and Context

1. Clarity and accessibility of the text

a. Major

- i. Where the term “lesion” has been used to describe an enhancer region is incorrect terminology.
- ii. In Figure 2 and the referential text, the authors use BM ILC2 and ILC2P interchangeably which is confusing because (a) they differentiate these cells from each other in Figure 3 and (b) the distinction between these two cells has been described before. ICOS expression represents one of the earliest surface markers of ILC2 differentiation from the ILCP, whereas ST2 and CD25 are markers of more differentiated ILC2 populations (Hoyler 2012 Immunity; Constantinides 2014 Nature; Ishizuka 2016 Nat. Imm.; Yu 2016 Nature; Kasal 2021 JEM). Therefore, the authors should refer to BM ILC2s in Figure 2 as LSIG cells (Hoyler 2012) or iILC2s (Constantinides 2014 Nature) and refer to ILC2P as cells that are ST2- CD25- ICOS+ but still a4b7+, which marks ILC precursors in the BM (Hoyler 2012 Immunity; Constantinides 2014 Nature; Yang 2015 Nat. Imm.; Ishizuka 2016 Nat. Imm.; Harly 2018 JEM; Kasal 2021 JEM)

b. Minor

- i. Authors accidentally say “IL-17RB+ PD-1+ late ILC2-committed precursors” instead of “IL-17RB+ PD-1- late ILC2-committed precursors” in the second to last sentence of the Introduction.
- ii. Supp. Fig. 1c is referred to in the second paragraph of the Results section where Supp. Fig. 1d should be referenced.

iii. Fig. 4d is referred to in the seventh paragraph of the Results section where Fig. 3d should be referenced.

2. Consideration of previous work

a. Major

i. Insufficient discussion and credit are given to previous reports describing Gata3 enhancers involved in the immune system (Kasal 2021 PNAS; Ohmura 2016 JCI). Particularly, Kasal et. al. demonstrated that Gata3 +672/764 (SE1) is necessary for the elevated levels of GATA3 seen in mature ILC2s, which is a central focus of this manuscript.

ii. How the authors delineate stages of ILC2 development (early vs. late) needs to be detailed in the introduction with appropriate and sufficient reference to Yu 2016 Nature, Ishizuka 2016 Nat. Imm., Walker 2019 Immunity, and Xu 2019 Immunity.

b. Minor

i. CHILP as a distinct intermediate ILC precursor cell does not exist but is instead a mix of differentiated ILC precursor cells (Kasal 2021 JEM).

ii. Limited background information is given as to the function of Cnot6l or what is known of other GATA3 targets genes.

Significance

1. Major

a. The claim of a "novel regulatory mechanism" for Gata3 expression during ILC2 development is inappropriate as this has already been described before (Kasal 2021 PNAS). The data presented here on G3SEKO mice is congruent with data presented in Kasal 2021 PNAS, but only expands on the previous report in identifying the late-ILC2P in the BM and lung as the necessary point of G3SE function.

b. The finding that G3SE is responsible for the development of late-ILC2P and high Gata3 expression in late-ILC2Ps is not entirely novel, as Kasal 2021 PNAS demonstrated that Gata3 +672/764 (SE1) is required for ILC2P (early- and late-ILC2P) development.

2. Minor

a. Results describing impaired ILC2 development and function during allergic inflammation in G3SEKO mice are expected and have been demonstrated before in Kasal 2021 PNAS.

Suggested Improvements

1. Eliminate the usage of super enhancer as a descriptive term throughout the publication.

Alternatively, demonstrate that the described regions are indeed super enhancers. This can be accomplished via:

a. ChIP-seq of Mediator in ILC2s

b. Analysis of key ILC2 transcription factor binding in the regions using public or newly generated ChIP-seq data sets (i.e., GATA3, Bcl11b, ROR α)

c. Perturbation of the region via CRISPR/Cas9-mediated deletion to demonstrate robustness

2. Generate and characterize an individual deletion of SE2 to formally demonstrate the involvement of this region in regulating ILC2 development and Gata3 expression. It is possible that SE1 and SE2 act additively, synergistically, or sequentially to induce Gata3 expression. Deletion of SE2 will also expand the analysis of the Gata3 locus beyond what was determined in Kasal 2021 PNAS and here using G3SEKO mice.

3. Further characterization of the induction of Cnot6l by GATA3 and the involvement of Cnot6l in regulating ILC2 development. This can be accomplished via:

a. Retroviral overexpression of Cnot6l in G3SEKO mice, akin to the Gata3 overexpression in Fig. 2d.

b. Luciferase assay using the GATA3 binding region within the Cnot6l locus identified in Fig. 5e.

4. Re-phrase/re-evaluate interpretation that G3SE does not regulate Th2 cell differentiation or provide further evidence that Th2 cell differentiation is unimpacted. This can be accomplished via:

a. Evaluating Th2 cell differentiation in vivo using type 2 inflammation models (i.e., house dust mite, parasitic worm infection, OVA-alum challenge)

5. The findings herein can be enhanced by linking observations with findings in other papers (Constantinides 2014 Nature, Ishizuka 2016 Nat. Imm., Kasal 2021 JEM). To do so, the authors could assess the surface expression of ICOS and a4b7 in the ILCP, early-ILC2P, late-ILC2P, and iILC2 populations in the BM and lung.

a. Does ICOS+ PD-1+ ILCP accumulation account for the increased frequency of CD25- PD-1+ ILC precursors observed in Figure 2b?

6. Provide more discussion comparing the findings on G3SE with those in Kasal 2021 PNAS describing Gata3 +672/764. Where are the findings supported by previous work and in what ways do they expand our understanding of ILC2 development?

Responses to Reviewer's comments

Reviewer #1 T cell transcriptional regulation (Remarks to the Author):

In this study, the authors aimed to understand ILC2 development driven through controlled high expression of GATA3 via cell type specific super enhancers (SE). The authors generated an enhancer deleted mice that were missing 2 tandem GATA3 SE regions (G3SEKO) and evaluated the ILC2 phenotype. G3SEKO cells showed cell intrinsic impairment to become mature ILC2 in tissues, and loss of high GATA3 expression during transition from ILC precursor to mature ILC2 in vitro. In contrast, G3SEs seems dispensable for maintaining high GATA3 expression in ILCP in BM and lung. Single cell RNA-seq analysis of BM cells illustrated possible difference between WT and G2SEKO and guided the authors to identify Cnot6l as a potential downstream target of GATA3 in supporting a later stage of ILC2 maturation. When challenged to induce type 2 inflammation in lung, G3SEKO mice exhibited muted type 2 response due to the lack of ILC2 population in the tissue.

The work clearly demonstrated the importance and the critical time window of the GATA3 super enhancers to support fully developed ILC2 in the tissues.

We appreciate the careful reading and thoughtful comments on our manuscript.

However, the report suffers from the fact that scRNA-seq analyses are based on seriously under-sampled and WT/KO unbalanced data, rendering interpretation of computational analyses unconvincing.

Repeating scRNA-seq with more cell capture per genotype and confirming the reproducibility of observations based on WT-KO balanced samples is highly recommended.

We completely agree with the Reviewer's comment. Unfortunately, our first scRNA-seq samples were under-sampled due to the low numbers of α LP cells in G3SEKO bone marrow. We performed scRNA-seq again with abundant cells (WT: 5230 cells, G3SEKO: 1140 cells). The results were mostly similar to 1st scRNA-seq data, besides clustering numbers. Combined scRNA-seq data was presented in Fig. 3 and Fig. 5. Accordingly, we changed the entire paragraphs related to scRNA-seq of the Result

section (Page 8, 3rd paragraph-Page 9, 1st paragraph, Page 11 3rd paragraph-Page 13, 1st paragraph).

Also lacking is any validation of intended CRISPR deletions of the target regions, or evaluations of off target editing in their newly generated mouse line.

We agree with the Reviewer's comment. Validation of deletion for two lines was presented in Supplementary Fig. 2B. To analyze off-target editing, we first performed CAS-OFFinder with the parameters "mismatch number 2, DNA bulge size 1, RNA bulge size 1" to identify the candidates. We found 26 candidates for downstream crRNA and 7 candidates for upstream crRNA. T7 analyses were performed for mixed equimolar DNA of G3SEKO and C57BL/6 WT (not littermate). Results are shown in Supplementary Fig. 3.

Page 5, 3rd paragraph of the Result section, added as follows:

"We successfully obtained two lines, 37 bp deletion and 11 bp deletion (Supplementary Fig. 2B), which have no obvious off-target effect (Supplementary Fig. 3)."

Page 19, 2nd paragraph of the Method section, added as follows:

"To analyze off-target editing, we first perform CAS-OFFinder⁴⁷ with the parameters "mismatch number 2, DNA bulge size 1, RNA bulge size 1". We found 26 candidates for downstream crRNA and 7 candidates for upstream crRNA in G3SEKO mice. T7 analyses were performed for mixed equimolar DNA of G3SEKO and C57BL/6 WT mice (Supplementary Fig. 3)."

Overall, the report in its current form falls short of being fully developed strong story. However, a relatively straightforward addition of repeat data analyses as well as molecular diagnosis of DNA deletions (SE1 and SE2) will improve the quality of the paper.

We appreciate all the Reviewer's comments, and we believe our revision based on your suggestions surely strengthen this manuscript's clarity and confidence.

Reviewer #2 supervisor for reviewer #3

Reviewer #3 ILC2 (Remarks to the Author):

The authors described a newly identified super enhancer governing GATA3 expression specifically in ILC2s. Deletion of this region in vivo generates a mature ILC2 deficient mouse, and therefore does not mount an allergic response in the lung when exposed to allergens. Their work has revealed a novel layer of regulation for ILC2s which is complementary to other studies showing the requirement of GATA3 for ILC2 development and function. This is an interesting work, well executed and that will contribute to our understanding of the regulation of ILC2s.

I would like to put forward a few comments/questions for the authors:

First, we appreciate the Reviewer's constructive comments and suggestions. We also appreciate the opportunity to allow us to revise our manuscript.

1. Characterization of the G3SEKO mouse. GATA3 is likely to be expressed in other cells apart from ILCs. What is the distribution of other immune cell types in the tissues analyzed?

We appreciate the Reviewer's comment. We analyzed various immune cell types, including neutrophils, tissue-resident macrophages, T cells, B cells, and dendritic cells in the lungs and livers of WT and G3SEKO mice (n=5). Our findings indicate no apparent differences between WT and G3SEKO mice in these cell populations. We put the number of each population in the Supplementary Fig. 5A and 5B.

Page 6, 1st paragraph of the Result section, added as follows:

“However, **other immune cell populations in the lung and liver** (Supplementary Fig. 5A, B) and T cell development and GATA3 expression in the thymus (Supplementary Fig. 5C, D) were indistinguishable between G3SEKO and WT mice.”

2. Figure 1: Quantification of ILCs in the HDM model should be shown.

Upon the request of the Reviewer, we have included the absolute number of ILC2 and CD4 T cells in the HDM models in Supplementary Fig. 1B.

3. Figure 2. The authors conclude that ILCP CD25⁻ PD1⁺ cells accumulate in the BM of G3SEKO mice. Could the authors show absolute numbers?

We thank the Reviewer's comments. The accumulation of CD25⁻PD-1⁺ cells was observed in the in-vitro culture experiment in Fig. 2B. We cultured a few hundred BM ILCPs on the OP9-DL1 cells for five days and harvested entire cells for FACS analysis. Unfortunately, the cells were too low to show accurate absolute numbers. Instead, we include the absolute numbers of IL17RB⁻ ILCP and IL17RB⁺ early ILC2-committed precursors in the revised manuscript in Fig. 2A. Both populations and the PD-1⁺ ILCP of the previous manuscript did not accumulate in the BM.

4. Figure 3: Can the authors show gating strategy for the purification of αLPs in the scRNAseq experiment?

Upon the Reviewer's request, we have incorporated the gating strategy used to purify αLPs in Supplementary Fig. 8A.

Page 8, 3rd paragraph of the Result section, changed as follows:

“To further investigate the precise stage of ILC2 development that requires high GATA3 expression induced by G3SE, we performed single-cell RNA sequencing (scRNA-seq) for αLPs in WT and G3SEKO mice (Supplementary Fig. 8A).”

5. Figure 3: scRNAseq experiment (Figure 3d-e) does not reflect the accumulation of PD1⁺ ILC2Ps observed in Figure 2b.

We appreciate the Reviewer's comment. The plot of the corresponding cell populations of the scRNA-seq experiment is shown in Fig. 2A. In Fig. 2A, the accumulation of PD-1⁺ ILCPs was not observed. Fig. 2B shows the plot of cultured cell populations of ILCPs on OP1-DL1 cells for 5 days. New scRNA-seq data revealed that G3SEKO αLP accumulated in cluster 6, which might reflect PD-1⁻IL17RB⁺ late ILC2-committed precursors in Fig. 3I.

6. Figure 3: Why are there so few G3SEKO αLPs sequenced? Can this alter the DEG analysis downstream?

We completely agree with the Reviewer's comment. Unfortunately, our first scRNA-seq samples were under-sampled due to the low numbers of α LP cells in G3SEKO bone marrow. We performed scRNA-seq again with abundant cells (WT: 5230 cells, G3SEKO: 1140 cells). The results were mostly similar to 1st scRNA-seq data, besides clustering numbers. Combined scRNA-seq data was presented in Fig. 3 and Fig. 5. Accordingly, we changed the entire paragraphs related to scRNA-seq of the Result section (Page 8, 3rd paragraph-Page 9, 1st paragraph, Page 11 3rd paragraph-Page 13, 1st paragraph).

As per the Reviewer's expectation, DEGs were slightly changed, and a minimal percentage of the gene expression is lower in new scRNA-seq data because of the dilution of UMI per cell by the increased cell numbers. However, the same highly different genes were still observed in new scRNA-seq data. Surprisingly, integration analysis of new scRNA-seq data and new ATAC-seq data in Fig. 5 shed light on the same genes, *IL1rl1*, *Cnot6l*, *Ptpn13*, and *Inpp4b*. One surprising thing is that the newly identified target, *Slc7a8*, has recently been validated and shown to be involved in regulating ILC2 numbers and functions [Panda SK et al., 2022 PNAS], indicating the robustness of this approach.

7. Figure 3: The blockade in ILC2 development is in Il18ra+ ST2- ILC2Ps in the G3SEKO mouse. Analysis of the activation of the gata3 SEs is shown in figure 1 for lung ILC2s. Are the Gata3 SEs also active in Il18ra+ ST2- ILC2Ps?

We thank the Reviewer's comments. H3K27ac ChIP-seq experiments used in Fig. 1 are challenging for applying the ILC2 progenitors because of the low cell numbers. Therefore, we performed ATAC-seq for IL17RB⁻PD-1⁺ ILCPs, early and late ILC2-committed precursors, IL18R α ⁺ and IL18R α ⁻ST2⁺ ILC2s obtained from the bone marrow of WT mice, as well as IL17RB⁻ ILCPs and early and late ILC2-committed precursors obtained from the bone marrow of G3SEKO mice. Both SE1 and SE2 contained newly opened chromatin regions in late ILC2-committed precursors, which seemed to be opened even in early ILC2-committed precursors but not in IL17RB⁻ ILCPs. Substantial numbers of the accessible chromatin regions in IL17RB⁻ ILCPs were closed in late ILC2-committed precursors. We added these data in Supplementary Fig. 10 in the revised manuscript.

Page 10, 4th paragraph of the Result section, added as follows:

“To understand the activation status of SE1 and SE2 during ILC2 lineage development in the BM, we performed ATAC-seq for IL17RB⁻ ILCP, early and late ILC2-committed precursors, IL18Rα⁺ST2⁺ ILC2s, and IL18Rα⁻ST2⁺ ILC2s. Both SE1 and SE2 contained opened chromatin regions in late ILC2-committed precursors, which are already opened in early ILC2-committed precursors but not in IL17RB⁻PD-1⁺ ILCPs (Supplementary Fig. 10, purple triangles). A substantial number of open chromatin regions in IL17RB⁻ ILCPs were closed in late ILC2-committed precursors (Supplementary Fig. 10, blue triangles).”

8. Figure 4. Can the authors show absolute counts for the data presented in this figure.

Upon the Reviewer’s request, we have added the absolute numbers in Fig. 4.

9. Figure 4c. Staining for Il17rb is dim. The increase in IL17rb+ cells seems to be due to the increase in the double negative population. What are Il17rb- Pd1- cells?

We agree with the Reviewer’s comment. Indeed, the question of what the IL17RB⁻PD-1⁻ cells are is unknown as there is no established marker continuously expressed from the early ILC2-committed precursor stage to the mature ILC2 stage. However, our data suggest that ILC1 or ILC3 are not the main IL17RB⁻PD-1⁻ cells, as only 14% and 8% of IL18Rα⁺ST2⁻ cells in G3SEKO mice were Tbet⁺ cells and RORγt⁺ cells, respectively, as shown in Supplementary Fig 4B. We found that Sca-1 is not expressed in PD-1⁺ cells in the lung and BM but starts to express in late ILC2-committed precursors in the BM and IL18Rα⁺ST2⁻ cells in the lung. Indeed, almost all IL18Rα⁺ST2⁺ and IL18Rα⁻ST2⁺ cells in the lung and ST2⁺ cells in the BM express Sca-1. Therefore, Sca-1 might help trace ILC2 lineage from late ILC2-committed precursors.

In the lung, 80% of IL18Rα⁺ST2⁻ cells in G3SEKO mice express Sca-1 compared to 45% in WT mice. These data imply that ILC2-committed precursors are accumulated in IL18Rα⁺ST2⁻ population, presumably in IL17RB⁺PD-1⁻ and IL17RB⁻PD-1⁻ fractions.

Page 11, 2nd paragraph of the Result section, added as follows:

“Another ILC2 marker, Sca-1^{15,26}, started to express in ST2⁻IL18Rα⁺ cells, with subsequent upregulation during ILC2 development in WT mice (Supplementary Fig. 11A). Notably, in G3SEKO mice, the majority of accumulated ST2⁻IL18Rα⁺ cells expressed Sca-1 (Supplementary Fig. 11B).”

10. In Figure 3, ILC2 progenitors are analysed using Il17rb and Pd1. In figure 4 in the lung, ILC2 progenitors are gated first using Il18r and ST2 markers and then data is shown for IL17rb and Pd1 (Figure 4c). In figure 5, both lung and BM are analyzed in the same manner as IL18r+ ST2- ILC2P. Keeping the same gating strategy for ILC2P analysis would aid the understanding of the paper.

We thank the Reviewer's comments that keeping the same gating strategy for ILC2P analysis would aid the understanding of the manuscript. Upon the Reviewer's request, we have unified our gating strategy where we first analyzed ST2 vs. IL18R α staining to segregate ST2⁺ ILC2 cells from ILCP to preILC2 stage cells in Fig. 3, 4, and 5.

Subsequently, IL18R α ⁺ST2⁻ cells were analyzed by PD-1 vs. IL17RB staining in Fig. 3 and 5.

On the other hand, in Fig. 2, we will keep the PD-1 vs. ST2 staining for two reasons. One reason is that PD-1 and Zbtb16, but not IL18R α , are well-established markers for ILCP. Another reason is that IL18R α starts to be focused based on scRNA-seq data.

11. Figure 5: ST2+ ILC2s are reduced in shCnotl6 condition. Can the authors show Gata3 expression in shCnotl6 vs shControl, to exclude the possibility of ST2 expression downregulation? Can the authors show absolute numbers?

We thank the Reviewer's comments. Upon the request to demonstrate Gata3 expression in shCnot6l conditions compared to shControl, we added the plots in Supplementary Fig. 12D. The expression levels of Gata3 were not significantly different between shControl- and shCnot6l-transduced ST2⁺ cells and those transduced IL18R α ⁺ST2⁻ cells.

Regarding Fig. 5, we realized that the cell numbers were significantly influenced by the transduction efficiency of sh-vectors, which ranged from 2% to 15%. Consequently, we showed the data as the percentage of hNGFR⁺ cells rather than the percentage of CD45⁺ cells or absolute numbers.

Page 13, 2nd paragraph of the Result section, changed as follows:

“The proportion of ST2⁺ ILC2s was significantly decreased in the lung and BM cells derived from *Cnot6l*-targeted sh-RNA-transduced cells **without affecting GATA3 expression** (Fig. 5H, I, **Supplementary Fig. 12D**).”

12. Figure 5: Cnot6l downregulation does not seem to replicate G3SEKO phenotype at the level of IL18Ra+ ST2- progenitors, as it is not increased. Showing absolute numbers may clarify this for the reader.

We thank the Reviewer's comments.

First, our data imply that the Cnot6l-enhancer region becomes open during ILC2 development but is closed in ILCP, and the induction of Cnot6l by GATA3 through SE occurs in the ILC2-committed precursor stage. On the other hand, the GATA3-binding promoter region of Cnot6l is already opened in the ILCP stage (Fig. 5E), and Cnot6l is already moderately expressed at the ILCP stage (Supplementary Fig. 12C). Since sh-Cnot6l could reduce Cnot6l expression in a stage-independent manner, sh-Cnot6l-transduced cells seem not to replicate the G3SEKO phenotype. It is anticipated that mice lacking Cnot6l-enhancer region or conditional Cnot6l knockout mice functioning in IL17RB⁺ cells may replicate the phenotype of G3SEKO mice, which we are currently addressing in the next project.

Second, as mentioned above, showing the absolute numbers of sh-Cnot6l could be inappropriate due to the difference in the transduction efficiency of sh-vectors.

Page 18, 2nd paragraph of the Discussion section, changed as follows:

“To understand the precise function of CNOT6L in ILC2 development, further detailed analyses of mice lacking GATA3-binding sites in the *Cnot6l* region or IL17RB⁺ cell-specific conditional *Cnot6l* knockout mice are necessary.”

13. Figure 5: Are there any differences in lung or BM IL17rb- PD1+ ILC2s in the absence of Cnot6l?

We appreciate the Reviewer's comment. Unfortunately, we could not answer this question because we did not observe analyzable numbers of sh-RNA transduced cells in the fraction of PD-1⁺ cells, including IL17RB⁺ and IL17RB⁻, in the BM and lung regardless of non-silencing or sh-Cnot6l.

14. Figure 6: Can the authors show absolute quantification for ILC numbers in both allergy models. Also, can the authors show histograms for IL-5 and IL-13 cytokine staining?

Upon the Reviewer's request, we added the data of the absolute counts and histograms

for IL-5- and IL-13-staining in Fig. 6.

We appreciate all the Reviewer's comments, and we believe your suggestions surely strengthen this manuscript's clarity and confidence.

Reviewer #4 ILC2 (Remarks to the Author):

First, we appreciate the careful reading and insightful comments on our manuscript.

Validity

1. Methodological approach, data quality and presentation

a. Major

i. An insufficient number of cells (53) was collected for G3SE knockout α LP scRNA-seq and must be expanded to improve data quality.

We completely agree with the Reviewer's comment. Unfortunately, our first scRNA-seq samples were under-sampled. This unbalance was due to the low numbers of α LP cells in G3SEKO bone marrow. We performed scRNA-seq again with abundant cells (WT: 5230 cells, G3SEKO: 1140 cells). The results were mostly similar to 1st scRNA-seq data, besides clustering numbers. Combined scRNA-seq data was presented in Fig. 3 and Fig. 5. Accordingly, we changed the entire paragraphs related to scRNA-seq of the Result section (Page 8, 3rd paragraph-Page 9, 1st paragraph, Page 11 3rd paragraph-Page 13, 1st paragraph).

ii. There is insufficient rationale given by the authors as to why SE1, SE2, and G3SE should be described as super-enhancers. If this term is to be used, the authors must demonstrate that the regions are “composed of clusters of active enhancers” and represent a region with characteristics of super enhancers as first described in Whyte 2015 Cell. This includes dense occupation by Mediator, enhancer size, transcription factor density and content, ability to activate transcription, and sensitivity to perturbation.

We appreciate the Reviewer's comments. We agree with the need for a more detailed explanation of our methodology for identifying super-enhancers. Upon the Reviewer's request, we have made the following improvements.

Supplementary Fig. 1C: We depicted the typical enhancers as black bars. Those enhancers are identified by the Homer findPeaks program with option “-style histone” for H3K27ac ChIP-seq and “-style factor” for deposit TF ChIP-seq data.

Supplementary Fig. 1D: We showed a super-enhancer detection plot of control ILC2, which identified SE1, SE2, and SE-*Gata3* as super-enhancers.

We also re-analyzed the deposited ChIP-seq experiments of GATA3 (GSE11187), Bcl11b (GSE131082), and RORa (GSE146743). While RORa ChIP-seq data only identified 2005 typical enhancers and 29 super-enhancers, GATA3 ChIP-seq data identified 56138 typical enhancers and 534 super-enhancers, and Bcl11b ChIP-seq data identified 49892 typical enhancers and 854 super-enhancers. Given this limitation, we utilized GATA3 ChIP-seq, Bcl11b ChIP-seq, and a combination of equal tags from GATA3 ChIP-seq and Bcl11b ChIP-seq as the basis for super-enhancer identification, emphasizing the master transcription factors. While the identified super-enhancers by master transcription factors may not perfectly align with SE1 and SE2, they exhibit a considerable overlap with the regions of SE1 and SE2. This observation further supports the importance of SE1 and SE2 in ILC2 development. We showed the typical enhancers and super-enhancers of GATA3 and Bcl11b in Supplementary Fig. 1C.

We hope these additional details and adjustments address the concerns raised by the Reviewer regarding the characterization of SE1, SE2, and G3SE as super-enhancers in our study.

b. Minor

i. The choice to perform ChIP-seq on total CD4+ T cells sorted from the lung following house dust mite challenge likely dilutes the true Th2 cell signal at SE1 and SE2 from their data in Figure 1A.

We agree with the Reviewer's comment that CD4+ T cells sorted from the lung of HDM-induced asthma models contain substantial numbers of non-Th2 cells, and these non-Th2 cells might dilute ChIP-seq signals at SE1 and SE2 in CD4+ T cells. We have modified the relevant phrases in the revised manuscript.

Page 5, 2nd paragraph of the Result section, changed as follows:

“These data indicate that G3SE might play an important role in GATA3 expression in lung ILC2s.”

Page 6, 2nd paragraph of the Result section, added as follows:

“Regarding Th2 cells, it has recently been reported that the SE1 region is opened in *N. brasiliensis*-activated lung Th2 cells^{27, 30}. Because CD4 T cells obtained from the lung in HDM-induced asthma models are not exclusively Th2 cells, the low activation of G3SE in lung CD4 T cells (Fig. 1A) might result from the dilution of Th2 cells by other CD4 T cell subsets.”

2. Analytical approach

- No issues were identified with the statistical analysis.

3. Data interpretation and conclusions

a. Major

i. The claim that G3SE functions partially through *Cnot6l* is not fully supported. The authors show that *Cnot6l* expression is decreased in G3SEKO late-ILC2P and that knockdown of *Cnot6l* also impairs ILC2 development. Nevertheless, without further analysis, such as overexpressing *Cnot6l* in G3SEKO cells, the authors cannot make the claim.

In response to the Reviewer’s suggestion, we conducted *Cnot6l* overexpression experiments in G3SEKO ILC2 progenitors (ILCP) using the ILCP culture system employed for *Gata3* overexpression in Fig. 2D. However, unfortunately, our attempts did not significantly recover CD25⁺PD-1⁻ cell development from *Cnot6l*-overexpressed G3SEKO ILCP (data not shown).

Meanwhile, we performed new ATAC-seq analyses on IL17RB-PD-1⁺ ILCPs, IL17RB⁺PD-1⁺ early ILC2-committed precursors, IL17RB⁺PD-1⁻ST2⁻ late ILC2-committed precursors, IL18R α ⁺ST2⁺ ILC2s, and IL18R α ⁻ST2⁺ ILC2s in response to another Reviewer’s request. The newly opened enhancer in the *Cnot6l* region, shown in new Fig. 5E, starts to open in WT IL17RB⁺PD-1⁻ cells but not in G3SEKO IL17RB⁺PD-1⁻ cells, suggesting that GATA3 induced by G3SE might activate this enhancer.

Upon the Reviewer’s suggestion in Suggested Improvements 3b, we performed a luciferase assay for this enhancer. The enhancer activity was almost 3-fold increased by GATA3 induction, supporting the hypothesis that GATA3 induces *Cnot6l* using this enhancer after the IL17RB⁺PD1⁻ stage. We added luciferase assay data in Supplementary Fig. 12F. We believe these data support the hypothesis that “G3SE functions partially through *Cnot6l*”. On the other hand, we agree that further

experiments are needed to confirm the hypothesis, and this limitation is mentioned in the Discussion in the revised manuscript.

Page 13, 2nd paragraph of the Result section, added as follows:

“The *Cnot6l*-enhancer region, identified by ATAC-seq (Fig. 5E), was accessible during ILC2 development in WT mice but maintained a closed conformation in G3SEKO ILCs (Fig. 5E). We also conducted a luciferase assay targeting the *Cnot6l*-enhancer region and found that the forced expression of GATA3 increased the *Cnot6l*-enhancer activity (Supplementary Fig. 12F).”

Page 18, 2nd paragraph of the Discussion section, added as follows:

“The enhancer region in *Cnot6l* locus became accessible in the late ILC2-committed precursors in WT mice but remained closed in G3SEKO mice (Fig. 5E). Furthermore, GATA3 overexpression increased the enhancer activities (Supplementary Fig. 12F). These results indicate that GATA3 induction by G3SE results in the elevation of *Cnot6l* in the late ILC2-committed precursors.”

Page 18, 2nd paragraph of the Discussion section, changed as follows:

“To understand the precise function of CNOT6L in ILC2 development, further detailed analyses of mice lacking GATA3-binding sites in the *Cnot6l* region **or conditional *Cnot6l* knockout mice functioning in IL17RB⁺ cells** are necessary.”

Page 23, 2nd paragraph of the Method section, added as follows:

“Luciferase reporter assay

Luciferase reporter assay was performed as previously described with minor modifications⁵¹. *Cnot6l* enhancer region (chr5:96111540-96112644) was cloned into pGL4.23[luc2 minP] vector (Promega). 293T cells were transfected with either pGL4.23-empty-enhancer vectors or pGL4.23-*Cnot6l*-enhancer vectors and either MSCV-empty-ires-Thy1.1 vectors or MSCV-Gata3-ires-Thy1.1 vectors in the presence of pGL4.74-[hRluc TK] vectors. Twenty-four hours later, RLUs were assessed with a dual luciferase assay system (Promega). The enhancer activity was normalized by the mean of RLU signals of MSCV-empty-IRES-Thy1.1 vector-transduced conditions.”

ii. The authors state “Although we did not confirm the respective role of the two tandem SEs, they are more likely to function together ... [Gata3 +672/764^{Δ/Δ}] was almost identical to our SE1 region (678–784 kb); therefore, the markedly strong

reduction in G3SEKO mice is due to the combination of SE1 and SE2 regions. Thus, SE1 and SE2 additively contribute to the GATA3 induction required for ILC2 development. Without deleting SE2 separately from G3SE, the authors can make no substantive claim as to the function of SE2 and the difference between G3SEKO and Gata3 +672/764 (SE1) KO mice.

Upon the Reviewer's request, we generated mice lacking SE2. Surprisingly, G3SE2KO mice demonstrated a reduction of ILC2s in the lung and liver but not in the BM. Consistently, GATA3 expression was reduced in the lung and liver ILC2s but not in the BM ILC2 lineage. On the contrary, Kazal's GATA3 +672/762^{Δ/Δ} mice demonstrated a more significant impact on BM ILC2s than ILC2s in the lung and small intestine. These results suggest that SE1 and SE2 have different roles in GATA3 expression, in which SE1 is vital for earlier differentiation in the BM, but SE2 is vital in the peripheral tissues. Furthermore, the phenotype of single super-enhancer deletion of SE1 or SE2 was weaker than that of G3SEKO mice, especially in the BM, suggesting that SE1 and SE2 might compensate for each other. We have shown these data in the revised manuscript.

Page 14, 3rd paragraph of the Result section, added as follows:

“Mice lacking the SE2 region show reduced ILC2s in peripheral tissues but not in BM.

It has recently been reported that GATA3 +672/762^{Δ/Δ} mice exhibit a 75% reduction of lung ILC2s, 68% of small intestinal ILC2s, and 89 % of BM ILC2s²⁷, suggesting that the SE1 region have more significant impact on ILC2s in BM than peripheral tissues. In contrast, G3SEKO mice showed almost complete deficiency of ILC2s in the BM and peripheral tissues. To further our understanding of the roles of SE1 and SE2 regions during ILC2 development, we generated mice lacking the SE2 region (G3SE2KO mice, Fig. 7A). G3SE2KO mice showed reduced numbers of lung and liver ILC2s by 67% and 58%, respectively (Fig. 7B, C). In contrast, G3SE2KO mice showed normal ST2⁺ ILC2 development in the BM (Fig. 7D). The development of liver ILC1s and ILC3s was not affected by the absence of the SE2 region (Fig. 7C). Consistently, GATA3 expression in the lung and liver ILC2s was modestly decreased, while GATA3 expression in ST2⁺ ILCs in the BM was almost normal in G3SE2KO mice (Fig. 7E-G). These results indicate that SE2 is pivotal in ILC2 development and GATA3 expression in peripheral tissues.”

Page 17, 2nd and 3rd paragraphs of the Discussion section, added as follows:

“The impacts of the *Gata3* enhancer regions on immune systems have been intensively investigated^{24,27}. For instance, the enhancer region located 278-285 kb downstream from *Gata3* is reported to play crucial roles in CD4 T cell development²⁴ and ILC1 and ILC2 development²⁷. However, the lack of a 278-285 kb region did not affect the expression of GATA3 and functions of mature ILC2s²⁷. On the other hand, *Gata3* +674/762^{ΔΔ} mice, which lack almost identical region to the SE1 region (678–784 kb), showed reduced ILC2 numbers and reduced GATA3 levels in ILC2s in both BM and peripheral tissues. *Gata3* +674/762^{ΔΔ} mice also showed impaired ILC2-dependent allergic inflammation²⁷. However, the reduction of ILC2s in *Gata3* +674/762^{ΔΔ} mice is incomplete. In this study, we showed that G3SEKO mice exhibited a complete absence of ILC2s in the BM and peripheral tissues. We also demonstrated that G3SE2KO mice showed reduced ILC2 numbers and reduced GATA3 levels in ILC2s in the peripheral tissues but maintained intact ILC2 development in the BM. These findings suggest that SE1 and SE2 act in concert to develop ILC2 in peripheral tissues and that SE1 but not SE2 plays pivotal roles in ILC2 development in the BM. This notion is consistent with a recent study employing high-resolution chromosomal conformation capture, showing that the regions identical to SE1 and SE2 bound to the *Gata3* TSS in ILC2s but not in ILC1s or ILC3s⁴⁶.

Regarding the SE1 region, Kasal et al. subdivided *Gata3* +674/762 region into *Gata3* +674/710 and +710/762 regions and generated mice deficient in each region. They found that *Gata3* +710/762^{ΔΔ} mice showed only a mild reduction of ILC2 number in the lung, and *Gata3* +674/710^{ΔΔ} mice showed normal ILC2 number in the lung, indicating that *Gata3* +674/710 and +710/762 regions redundantly induce the development of ILC2s in the lung²⁷. Together with our findings, G3SE contains multiple enhancer regions contributing to ILC2 development and the deletion of the entire G3SE region results in the complete absence of ILC2s in the BM and peripheral tissues.”

Page 19, 2nd paragraph of the Method section, changed as follows:

“To form ribonucleoprotein complexes (RNPs), the gRNAs flanking the G3SE region (chr2; 9101999-9102118 and chr2; 9402399-9402518, Supplementary Table 1 and Supplementary Fig. 2) and G3SE2 region (chr2:9283835-9283857 and chr2:9402453-9402475, Supplementary Table 1 and Fig. 7A) were annealed with tracrRNA (IDT) and bound to Cas9 protein (Thermo Fisher Scientific).”

b. Minor

i. The claim of no Th2 cell deficiency is overstated. In vitro differentiation of Th2 cells is insufficient to demonstrate no impact on Th2 cell development particularly

given that Kasal 2021 PNAS demonstrated that Gata3 +672/764 (SE1) partially controls Gata3 expression in Th2 cells and is active in Th2 cells.

Upon the Reviewer's comments, the description regarding Th2 cells was weakened in the revised manuscript. In addition, experiments with OVA-induced asthma models were conducted using the same method as Kasal et al. reported (Supplementary Fig. 6).

Page 2, in Abstract, changed as follows:

“G3SE-deficient mice exhibit ILC2 deficiency in the bone marrow, lung, liver, and small intestine **with minimal impact on** other ILC lineages or Th2 cells.”

Page 6, 2nd paragraph of the Result section, added as follows:

“Therefore, we assessed in vivo Th2 cell development and function using ovalbumin (OVA)-induced asthma models (Supplementary Fig. 6A), which are known to be less influenced by ILC2 deficiency. Notably, the numbers of eosinophils, Th2 cells (defined by CD3 ϵ ⁺CD4⁺ST2⁺ T cells), and IL-5- or IL-13-producing Th2 cells were comparable between WT and G3SEKO mice (Supplementary Fig. 6B-E), while GATA3 expression in Th2 cells in bronchoalveolar lavage fluid (BALF) was slightly decreased in G3SEKO mice (Supplementary Fig. 6F). These results suggest that the G3SE region is critical in ILC2 development but only partially impacts Th2 cell development.”

Page 20, 3rd paragraph of the Method section, added as follows:

“For ovalbumin (OVA)-induced allergic airway inflammation, mice were immunized intraperitoneally (i.p.) with 50 μ g OVA absorbed in aluminum hydroxide gel (Thermo Fisher Scientific, Waltham, MA) on day 0, 7, and 14 and intranasally challenged with 100 μ g OVA on day 20, 21, and 22. Twenty-four hours after the last challenge, BALF and lungs were harvested.”

ii. The finding from ATAC-seq data that SE1 is inaccessible in Th2 cells is incongruent with re-analysis of public ATAC-seq data in Kasal 2021 PNAS (Data from Shih 2016 Cell on ILC2s and Th2 cells sorted from active parasitic worm infection). Moreover, the authors own ChIP-seq data for H3K27ac shows binding in SE1 and SE2 in CD4⁺ T cells sorted after house dust mite exposure but is likely reduced due to diluted Th2 cell signal among total CD4⁺ T cells.

We agree with the Reviewer's comment. We have changed the manuscript as above
Minor-i.

iii. Papain-induced allergic inflammation in the lung is IL-33 dependent (Halim 2014 Immunity). Therefore, the statement that papain would “exclude the possibility that attenuated eosinophilic inflammation was caused by the decreased expression of ST2” is incorrect.

We appreciate the Reviewer's suggestion. Halims et al. have shown that Il-33^{-/-} mice showed no signs of papain-induced inflammation. Therefore, our statement is incorrect, and we corrected this error in the revised manuscript.

Page 14, 2nd paragraph of the Result section, changed as follows:

“To further examine the role of G3SE *in vivo*, we also employed a papain-induced allergic airway inflammation model (Fig. 6E).”

Clarity and Context

1. Clarity and accessibility of the text

a. Major

i. Where the term “lesion” has been used to describe an enhancer region is incorrect terminology.

We thank the Reviewer for careful reading. We corrected them.

ii. In Figure 2 and the referential text, the authors use BM ILC2 and ILC2P interchangeably which is confusing because (a) they differentiate these cells from each other in Figure 3 and (b) the distinction between these two cells has been described before.

We thank the Reviewer for your insightful comments regarding the terminology used in ILC2 lineages in the BM. First, we want to clarify that we are not confused regarding the use of “ILC2P”, “BM ILC2”, and “ILC2-committed precursors” in the main text in the previous version of the manuscript. Since BM ILC2 is called ILC2P in many papers [Tsou & Artis, Nature 2022, Jarick & Klose, Nature 2022, Huang & Germain Nature 2022, Zhong & Zhu Immunity 2022, Walker & McKenzie, Immunity 2019, Harly & Bhandoola, JEM 2018], we explicitly state that we refer to BM ILC2 when we are

referring to cells known as ILC2P. We have consistently spelled it out in the revised manuscript as “ILC2-committed precursors” to avoid confusion with LSIG-referred ILC2P.

ICOS expression represents one of the earliest surface markers of ILC2 differentiation from the ILCP, whereas ST2 and CD25 are markers of more differentiated ILC2 populations (Hoyler 2012 Immunity; Constantinides 2014 Nature; Ishizuka 2016 Nat. Imm.; Yu 2016 Nature; Kasal 2021 JEM). Therefore, the authors should refer to BM ILC2s in Figure 2 as LSIG cells (Hoyler 2012) or iILC2s (Constantinides 2014 Nature) and refer to ILC2P as cells that are ST2-CD25- ICOS+ but still a4b7+, which marks ILC precursors in the BM (Hoyler 2012 Immunity; Constantinides 2014 Nature; Yang 2015 Nat. Imm.; Ishizuka 2016 Nat. Imm.; Harly 2018 JEM; Kasal 2021 JEM)

We agree with the importance of ICOS in the field of ILC2 progenitor in the BM. First, we would like to explain why we used IL17RB instead of ICOS to trace ILC2 lineages. Ishizuka et al. showed that PLZF⁺ICOS^{hi} cells tend to develop into ILC2 lineages, but these cells still have multipotency and show triple or double expression of Tbet, ROR γ t, and GATA3 [Fig. 7 in Ishizuka 2016 Nat Immunol]. On the other hand, Yu et al. showed that Bcl11b⁺IL17RB⁺PD-1⁺ cells develop into only ILC2s [Extended Data Figure 8b and 8c in Yu 2016 Nature]. These results indicate that IL17RB is a better marker than ICOS for the ILC2-committed precursors. However, we found that ICOS^{hi}PD-1⁺ cells and IL17RB⁺PD-1⁺ cells are almost identical (below Figure A). Meanwhile, we realized that IL17RB has a clear threshold, while ICOS^{hi} and ICOS^{int} are sequential. Therefore, sorted ICOS^{hi} cells might include ICOS^{int} cells, which may explain the discrepancy in these papers’ data. So, we partially disagree with the

Reviewer's comment that ILC2-committed precursors should be defined by "ICOS^{hi}" instead of "IL17RB⁺".

In addition, we have another reason to use IL17RB instead of ICOS. As described above, the gating of ICOS^{hi} and IL17RB⁺ indicated almost the same cells in PD-1⁺ cells. However, when considering late ILC2-committed precursors (PD-1⁻ cells), these gating did not indicate the same cells.

As shown in Figure B above, regardless of LPAM-1 positivity, ST2⁺ cells are IL17RB⁺ and ICOS^{int}. PD-1⁻ST2⁻ fraction contains three fractions: IL17RB⁺ICOS^{int} (similar IL17RB/ICOS levels to ST2⁺ cells), IL17RB⁻ICOS^{int} (similar ICOS levels to ST2⁺ cells), and IL17RB⁻ICOS⁻ cells, especially under LPAM-1^{low} gating.

As shown in Figure C above, IL17RB⁺ICOS^{int} cells (green rectangle) developed into CD25⁺ICOS^{hi} cells in OP9-DL1 culture systems, while IL17RB⁻ICOS^{int} cells (yellow rectangle) did not develop into ILC2s but into CD25^{hi}ICOS^{low} cells.

We want to use the same markers for detecting early and late ILC2-committed precursors. If we choose ICOS for the ILC2 lineage-tracing marker, ICOS^{int} cells include not only ST2⁺ and late ILC2-committed precursors but also IL17RB⁻ICOS^{int} cells in the PD-1^{low} population. Therefore, we chose IL17RB to evaluate early and late ILC2-committed precursors instead of ICOS. Because these things are too complicated and not the main story of our manuscript, we only showed the above results to the Editor and Reviewers.

Second, we explain why we stopped using LPAM-1 gating to identify late ILC2 committed precursors in Fig. 3I. We agree with the use of LPAM-1⁺ gating for detecting ILCPs because all IL17RB⁻ ILCPs express high levels of LPAM-1 [Constantinides 2012 Nature; Supplementary Fig. 9A]. However, many reports demonstrated (but did not mention) that ILC2s in the BM contained substantial proportions of LPAM-1⁻ cells [Hoyler 2012 Immunity; Supplementary Fig. 9A]. Because LPAM-1 is recognized as a maturation marker of ILC2 [Hoyler 2012 Immunity], LPAM-1⁻ cells might be excluded in the context of ILC2 progenitors. However, as shown in Supplementary Fig 9A-C, LPAM-1 was downregulated in IL17RB⁺PD-1⁺ cells. These findings indicate that we should re-consider the use of LPAM-1 gating to trace the later stage of IL17RB⁺PD-1⁺ cells. Indeed, ST2⁻IL17RB⁺PD-1⁻ cells were mainly observed in LPAM-1⁻ fraction (Figure B). These cells expressed high GATA3 at similar levels to ST2⁺ ILC2s (Fig. 3J) and developed into ILC2s in the OP9-DL1 culture (Supplementary Fig. 9D and Figure C). Therefore, the ILC2 precursors in the BM are not always LPAM-1⁺.

Third, we would like to explain why we use ST2 instead of CD25 for the ILC2 marker. We agree with the Reviewer's comment "ST2 and CD25 are markers of more differentiated ILC2 populations". Many papers defined ILC2 using CD25 or ST2. When we consider the just pre-ILC2 stage (late ILC2-committed precursors), the marker for defining ILC2s is important because CD25 and ST2 expressions do not start simultaneously. As shown in Fig. 7C, there are a substantial number of CD25⁻ cells in ST2⁺ cells. These CD25⁻ST2⁺ cells were recognized as the cells at the pre-ILC2 stage when we used CD25 as the marker for ILC2s (Supplementary Fig.7B, lower panels), while they are recognized as ILC2s using ST2 as the marker for ILC2s (Supplementary Fig. 7B, upper panels). Most CD25⁻ICOS⁺ cells, called ILC2P by Kazal's JEM paper, expressed ST2 (Supplementary Fig. 7B). Therefore, the marker we used to define ILC2s is also important. We considered the difference between ST2⁺ cells and ST2⁻ cells to be more fundamental than the difference between CD25⁺ cells and CD25⁻ cells (including ST2⁺CD25⁻ cells). Therefore, we selected ST2 as the marker for more differentiated ILC2s, and late ILC2-committed precursors should be ST2⁻ cells.

Fourth, we discuss the name of more differentiated ILC2 in the BM. Hoyler et al. called LSIG "ILC2P (lineage-specified precursors to ILC2s)" after Fig. 5 in their paper [Hoyler 2012 Immunity], and the same group referred to these cells as ILC2P instead of LSIG [Klose 2014 Cell]. Therefore, in many papers, ILC2s in the BM were called "ILC2P" with the citation of Hoyler 2012 Immunity. However, as the Reviewer is

concerned, this name is confusing in our manuscript because we focus on ILC2-committed precursors, which is a stage before ILC2. Therefore, we would not like to use “ILC2P”. The term “LSIG” has not been used in recent papers. Because we did not use Sca1, Id2, and Gata3 as markers, the term “LSIG” is inappropriate. “iILC2” is also used to refer to “inflammatory ILC2”. Recently, despite of low production of IL-5 and IL-13, ILC2s in BM have been shown to play the specific roles, indicating that they are not simply immature [Momiuchi 2021 Int Immunol; Sudo 2021 J Ex Med]. Therefore, the term “iILC2” is also confusing and inappropriate. Collectively, we would like to clear these two confusing issues, “the marker of ILC2” and “the name of ILC2”, by using ST2 as the definition of ILC2s and using the term “ST2⁺ ILC2s”, which refers to ILC2 in the BM. We understand that these things are still confusing. So, we put the terminology used in this manuscript in Supplementary Fig. 7A.

Page 7, 2nd paragraph of the Result section, added as follows:

“To investigate the specific developmental stage regulated by G3SE, we first explain the terms and definition of the ILC2 development stage used in this study (Supplementary Fig. 7A) because in the BM, ILC2 cells have been referred to by various names such as LSIG¹⁵, ILC2P^{13, 14, 15, 22}, iILC2⁹, and BM ILC2^{12, 16, 17}. In this study, we defined ILC2s in the BM as PD-1⁻ST2⁺ cells, terming them ST2⁺ ILC2s. In addition, cells already committed to ILC2 lineages but not yet exhibiting the ST2-positive stage (named as ILC2-committed precursors) have been recently identified as *Bcl11b*⁺IL17RB⁺PD-1⁺ cells^{10, 12} or ICOS^{hi}*Zbtb16*⁺ cells¹¹ among PD-1⁺ ILCPs. CD25⁻*Bcl11b*⁺*Zbtb16*⁻ cells and CD25⁻ICOS^{hi}*Zbtb16*⁻ cells were also identified as ILC2-committed precursors^{12, 14}. In this study, we distinguished ILC2-committed precursors by PD-1 expression: namely, we termed PD-1⁺ cells^{10, 11, 12} closer to ILCPs as early ILC2-committed precursors and PD-1⁻ cells^{12, 14} closer to ILC2s as late ILC2-committed precursors. Moreover, ILCPs were defined as PD-1⁺ST2⁻IL17RB⁻ cells and referred to as IL17RB⁻ ILCPs for excluding early ILC2-committed precursors (Supplementary Fig. 7A).”

Page 7, 3rd paragraph of the Result section, added as follows:

“Notably, late ILC2-committed precursors were scarcely observed within α LP when ST2 was used as an ILC2 marker (Supplementary Fig. 7B), which is a contrast to the findings using CD25 as an ILC2 marker¹⁴. The discrepancy can be attributed to using ST2 as an ILC2 marker instead of CD25 because approximately 5% of ST2⁺ ILC2s are CD25-negative (Supplementary Fig. 7C)”

Page 9, 2nd paragraph of the Result section, added as follows:

“To trace the development of ILC2 lineages, we questioned the use of LPAM-1⁺ gating for ILC2 lineage-tracing because of the presence of LPAM-1⁻ST2⁺ ILC2s¹⁵ (Supplementary Fig. 9A). IL17RB⁻ ILCPs expressed high levels of LPAM-1, and their expression reduced along with IL17RB induction (Supplementary Fig. 9A, B). ST2⁻ IL17RB⁺ cells, representing putative early and late ILC2-committed precursors, sustained low levels of LPAM-1 during PD-1 downregulation (Supplementary Fig. 9C). To examine the differentiation potential of IL17RB⁺PD-1⁻ cells, we cultured these cells on OP9-DL1 cells. Five days later, IL17RB⁺PD-1⁻ cells developed mostly into CD25⁺ICOS^{hi} ILC2s (Supplementary Fig. 9D). Collectively, we defined late ILC2-committed precursors as Lin⁻CD127⁺CD135⁻ST2⁻IL18R α ⁺PD-1⁻IL17RB⁺ cells. This observation aligns with the results obtained from scRNA-seq analysis.”

b. Minor

- i. Authors accidentally say “IL-17RB+ PD-1+ late ILC2-committed precursors” instead of “IL-17RB+ PD-1- late ILC2-committed precursors” in the second to last sentence of the Introduction.**
- ii. Supp. Fig. 1c is referred to in the second paragraph of the Results section where Supp. Fig. 1d should be referenced.**
- iii. Fig. 4d is referred to in the seventh paragraph of the Results section where Fig. 3d should be referenced.**

We thank the Reviewer for careful reading. We corrected them.

2. Consideration of previous work

1. Major

- i. Insufficient discussion and credit are given to previous reports describing Gata3 enhancers involved in the immune system (Kasal 2021 PNAS; Ohmura 2016 JCI). Particularly, Kasal et. al. demonstrated that Gata3 +672/764 (SE1) is necessary for the elevated levels of GATA3 seen in mature ILC2s, which is a central focus of this manuscript.**

We appreciate the Reviewer’s comments. We have discussed this point in more detail by referring to the previous papers, including Kasal 2021 PNAS, as described above.

ii. How the authors delineate stages of ILC2 development (early vs. late) needs to be detailed in the introduction with appropriate and sufficient reference to Yu 2016 Nature, Ishizuka 2016 Nat. Imm., Walker 2019 Immunity, and Xu 2019 Immunity.

We thank the Reviewer for the comments. We explained the details of ILC2 development in the BM in the revised manuscript. We also clarify how to delineate stages of ILC2 development in this manuscript as described above.

Page 3, 2nd paragraph of the Introduction section, added as follows:

“ILC2-lineage cells develop from ILC progenitors (ILCPs) that express PLZF (encoded by *Zbtb16*), PD-1, and LPAM-1 (Integrin $\alpha4\beta7$), but no ST2 or CD25 in bone marrow (BM)^{8, 9, 10}. ILCPs have the potential to develop into ILC1s, ILC2s, and ILC3s but not into NK cells or lymphoid tissue inducer (LTi) cells. Within the ILCP population, ILC2-committed precursors were identified using the expression of ICOS¹¹, *Bcl11b*^{10, 12, 13}, and IL17RB¹⁰ along with PD-1 and PLZF. The stage between *Zbtb16*⁺ cells, including ILC2-committed precursors, and ILC2s was recently identified. Kasal et al. reported that Lin⁻CD127⁺CD135⁻LPAM-1⁺Id2⁺CD25⁻ICOS⁺*Zbtb16*⁻ cells could differentiate into ILC2s¹⁴, while Xu et al. demonstrated that Lin⁻CD127⁺CD135⁻LPAM-1⁺Id2⁺CD25⁻ICOS⁻*Bcl11b*⁺*Zbtb16*⁻ cells predominantly differentiate into ILC2s¹². Hence, these cells were considered as the next stage of *Zbtb16*⁺ ILC2-committed precursors. Finally, ILC2s in BM express ST2 and CD25, serving as the source of tissue ILC2s^{8, 9, 10, 11, 15} and playing unique roles in BM under the stress conditions^{16, 17}.”

2. Minor

i. CHILP as a distinct intermediate ILC precursor cell does not exist but is instead a mix of differentiated ILC precursor cells (Kasal 2021 JEM).

We agree with Reviewer’s comment. We used PD-1⁻CD25⁻ fraction instead of the term “CHILP” in the revised manuscript.

Page 7, 3rd paragraph of the Result section, changed as follows:

“We first analyzed α -lymphoid progenitor (α LP) population (*Lin*⁻CD127⁺CD135⁻LPAM-1⁺ cells), which encompasses IL17RB⁻ ILCPs, early and late ILC2-committed precursors, ST2⁺ ILC2s, and PD-1⁻ST2⁻ cells (also known as CHILPs³², but now recognized as the mixture of iTiP and iILCP^{11, 14, 22}).”

Limited background information is given as to the function of Cnot6l or what is known of other GATA3 targets genes.

We thank the Reviewer for the comments. We added the background information of Cnot6l and other possible GATA3 target genes in the revised manuscript.

Page 13, 1st paragraph of the Result section, added as follows:

“*Slc7a8* is a transporter of arginine and large amino acids. *Il7ra^{cre}Slc7a8^{fl/fl}* mice have a reduced number of ILC2, and the rest of the ILC2s fail to produce IL-5 and IL-13³⁹. *Cnot6l* is one of the components of the CCR4-NOT complex, which functions as an mRNA deadenylase, and the CCR4-NOT complex is critical in early T cell development⁴⁰. *Ptpn13* is a tyrosine phosphatase that interacts with various molecules and regulates Th1 and Th2 differentiation through STAT4 signaling⁴¹. *Inpp4b* is a signaling protein that regulates the PI3K/Akt pathway through phosphatase activity, which hydrolyzes PI(3,4,5)P3 and PI(3,4)P2. It has recently been shown that *Inpp4b* directs ILC1 homing to cancer tissues using *NCR^{iCre}/Inpp4b^{fl/fl}* mice⁴². Because two of the five candidate genes, *Il1rl1* and *Slc7a8*, are reported as critical molecules for ILC2 development and functions, we hypothesize that the remaining three genes may also be crucial for ILC2 development.”

Significance

1. Major

a. The claim of a “novel regulatory mechanism” for Gata3 expression during ILC2 development is inappropriate as this has already been described before (Kasal 2021 PNAS). The data presented here on G3SEKO mice is congruent with data presented in Kasal 2021 PNAS, but only expands on the previous report in identifying the late-ILC2P in the BM and lung as the necessary point of G3SE function.

We appreciate the Reviewer’s comment. Upon the request of the Reviewer, we generated mice lacking SE2 and added considerable insight into the lineage-specific regulatory mechanism of Gata3 expression during ILC2 development. We have discussed this point in the Discussion section of the revised manuscript, as described above. We also removed “novel” from the revised manuscript.

Page 2, in Abstract, changed as follows:

“Our findings **uncovered a stage-specific** regulatory mechanism for GATA3 expression during ILC2 development.”

The finding that G3SE is responsible for the development of late-ILC2P and high Gata3 expression in late-ILC2Ps is not entirely novel, as Kasal 2021 PNAS demonstrated that Gata3 +672/764 (SE1) is required for ILC2P (early- and late-ILC2P) development.

We partially disagree with the Reviewer’s comment based on the terminology of ILC2P in Kasal’s PNAS paper. Kasal et al. used the term “ILC2P” referring to ICOS⁺Thy1.2⁺ cells pre-gated Lin⁻CD127⁺LPAM-1⁺ cells in Fig. 1D. Those ILC2P consist of 42.6% of Lin⁻CD127⁺LPAM-1⁺ cells in Fig. 1D and 65.4% of Lin⁻CD127⁺LPAM-1⁺ cells in Fig. S3B, suggesting that the majority of ILC2P in Kasal’s PNAS paper are ST2⁺ ILC2s. Therefore, we consider that Kasal et al. did not focus on early or late ILC2-committed precursors, and there is no evidence about the impact of Gata3 +672/764 on the ILC2-committed precursors in Kasal’s PNAS paper.

2. Minor

a. Results describing impaired ILC2 development and function during allergic inflammation in G3SEKO mice are expected and have been demonstrated before in Kasal 2021 PNAS.

We agree with the Reviewer’s comment. We added the following sentence in the revised manuscript.

Page 14, 1st paragraph of the Result section, changed as follows:

“Eosinophil numbers in the BALF and lung were significantly decreased in G3SEKO mice compared to those in WT mice (Fig. 6B, C), **consistent with a previous report of GATA3 +672/762^{Δ/Δ} mice, which lack a region nearly identical to the SE1 region²⁷.**”

Suggested Improvements

- 1. Eliminate the usage of super enhancer as a descriptive term throughout the publication. Alternatively, demonstrate that the described regions are indeed super enhancers. This can be accomplished via:**
 - a. ChIP-seq of Mediator in ILC2s**

- b. Analysis of key ILC2 transcription factor binding in the regions using public or newly generated ChIP-seq data sets (i.e., GATA3, Bcl11b, ROR α)**
- c. Perturbation of the region via CRISPR/Cas9-mediated deletion to demonstrate robustness**

We appreciate the Reviewer's suggestions. As mentioned above, we have added the methodology identifying super-enhancers and the data to support the term "super-enhancer" usage in the revised manuscript.

- 2. Generate and characterize an individual deletion of SE2 to formally demonstrate the involvement of this region in regulating ILC2 development and Gata3 expression. It is possible that SE1 and SE2 act additively, synergistically, or sequentially to induce Gata3 expression. Deletion of SE2 will also expand the analysis of the Gata3 locus beyond what was determined in Kasal 2021 PNAS and here using G3SEKO mice.**

Upon the Reviewer's request, we generated mice lacking SE2 to demonstrate the involvement of this region in regulating ILC2 development and Gata3 expression. We also discussed the function of SE1 and SE2 with the findings by Kasal et al. in the revised manuscript. We believe the analyses of mice lacking SE2 improve the significance of the manuscript.

- 3. Further characterization of the induction of Cnot6l by GATA3 and the involvement of Cnot6l in regulating ILC2 development. This can be accomplished via:**
 - a. Retroviral overexpression of Cnot6l in G3SEKO mice, akin to the Gata3 overexpression in Fig. 2d.**
 - b. Luciferase assay using the GATA3 binding region within the Cnot6l locus identified in Fig. 5e.**

We appreciate the Reviewer's suggestions. As mentioned above, although retroviral overexpression of Cnot6l did not yield the expected results, we obtained other data to support the role of G3SE for the induction of Cnot6l. We have added these data in the revised manuscript.

- 4. Re-phrase/re-evaluate interpretation that G3SE does not regulate Th2 cell differentiation or provide further evidence that Th2 cell differentiation is unimpacted. This can be accomplished via:**
- a. Evaluating Th2 cell differentiation in vivo using type 2 inflammation models (i.e., house dust mite, parasitic worm infection, OVA-alum challenge)**

We appreciate the Reviewer's suggestions. As mentioned above, we rephrase the interpretation regarding G3SE on Th2 cells. We also evaluated the roles of G3SE in Th2 cells in OVA-induced asthma models and added these data in the revised manuscript.

- 5. The findings herein can be enhanced by linking observations with findings in other papers (Constantinides 2014 Nature, Ishizuka 2016 Nat. Imm., Kasal 2021 JEM). To do so, the authors could assess the surface expression of ICOS and a4b7 in the ILCP, early-ILC2P, late-ILC2P, and iILC2 populations in the BM and lung.**

We appreciate the Reviewer's suggestions. We carefully analyzed ICOS and LPAM-1 expressions related to the previous papers as mentioned above. ICOS and LPAM-1 expression in BM ILC2 lineages were shown in above Figure B and Supplementary Fig. 9. Because ICOS and LPAM-1 expression of lung ILC2 lineages is not the main story of our manuscript, we only showed the results to the Editor and Reviewers.

- a. Does ICOS+ PD-1+ ILCP accumulation account for the increased frequency of CD25- PD-1+ ILC precursors observed in Figure 2b?**

We thank the Reviewer for the comments. The answer is yes. We assessed the expression of ICOS and LPAM-1 in the same experiment in Fig. 2B. as shown below fCD25⁻PD-1⁺ cells (marked with red circles) were present in G3SEKO ILCP-derived cells and exhibited intermediate levels of ICOS expression. The levels of LPAM-1 were low, like those observed in WT ICOS^{hi} PD-1^{low} ILC2s (marked with blue).

6. **Provide more discussion comparing the findings on G3SE with those in Kasal 2021 PNAS describing *Gata3* +672/764. Where are the findings supported by previous work and in what ways do they expand our understanding of ILC2 development?**

We appreciate the Reviewer's suggestions. Upon the Reviewer's request, we provided more discussion comparing the findings in Kasal's 2021 PNAS paper. We believe that this addition would expand our understanding of ILC2 development.

Again, we appreciate the Reviewer's comments, and we believe your suggestions surely strengthen this manuscript's clarity and confidence.

REVIEWERS' COMMENTS

Reviewer #1 (Remarks to the Author):

The authors have successfully responded to questions with new experiments and analysis. They also provided a balanced perspective on ILC2 development through in-depth understanding of previous literature that underlies their interpretation of ILC2 nomenclature. The revised manuscript offers a significant new insight into regulation of GATA3 expression during ILC2 development. I support it for publication in Nature Communication.

Reviewer #3 (Remarks to the Author):

This reviewer appreciate the efforts to respond to the questions raised during my revision of their work. Most of the issues have been satisfactorily addressed and contribute to a better understanding of the manuscript. The main conclusion of this work is related to the regulatory role of GATA3 enhancer in ILC2 development is well established. However, the functional data supporting the identification of Cnot6l as a mediator of GATA3SE-dependent ILC2 development is not the strongest part of the manuscript as:

- Reconstitution at week 5 does not allow for replenishment of ILC2s in peripheral tissues. Authors should wait a bit longer to analyze mice after BM transplant. Please refer to Jacquelot N., et al 2021.
- Evidence for efficient downregulation of Cnot6l in NGFR1+ cells in vivo is lacking.
- It is unclear how the quantification of ILC2s in transduced cells is done. Is it from a CD45+ NGFR1+ gate? Have the authors excluded Lineage positive cells or added any other ILC2 marker?
- Additionally, the authors could have tested the ability of Cnot6l overexpression to rescue the Gata3SEKO phenotype.
- In Supp. Fig. 12, shPtpn13 effects on ST2+ cells seem to be comparable to Cnot6l downregulation. However, Ptpn13 is regarded as not affecting development of ILC2s. This reviewer does not understand the criteria for choosing a gata3SE candidate based on the data shown. Is there a statistical test done on Supp. Fig 12E?

In conclusion, I would suggest to include Cnot6l functional data (Figure 5G,5h, and 5J) as supplementary information instead. Otherwise, I thank the authors for their work and look forward to see this manuscript published.

Reviewer #4 (Remarks to the Author):

In the revised version of their manuscript, authors Furuya et al. have substantively and significantly improved the quality of their submission. Most importantly, they have added to the significance of the manuscript through the addition of a knockout (G3SE2) that further subdivides their larger G2SE knockout and provides additional insight into the regulation of GATA3 in ILC2s beyond what has been reported in their manuscript and the literature. Furthermore, they have improved the impact of their manuscript through references to published literature, placing their findings in the context of the field at large, and through improvements in experimental rigour, including additional scRNA-seq data and further probing of Cnot6l. Lastly, they have improved the overall clarity of the manuscript through the addition of gating strategies and explanations of ILC2 development, as understood in the field. Overall the revisions the authors added have improved the impact of the paper and will be a welcome addition to the ILC2 and GATA3 field at large. Beyond the additional experiments proposed in the Discussion by the authors, it would be interesting to investigate the chromosomal conformation at the GATA3 locus in ILC2s and developing ILCs, particularly between G3SE and GATA3, and G3SE1 and G3SE2.

Reviewer #1 (Remarks to the Author):

The authors have successfully responded to questions with new experiments and analysis. They also provided a balanced perspective on ILC2 development through in-depth understanding of previous literature that underlies their interpretation of ILC2 nomenclature. The revised manuscript offers a significant new insight into regulation of GATA3 expression during ILC2 development. I support it for publication in Nature Communication.

Thank you for evaluating the revised manuscript. We are truly grateful for the insightful comments provided by the Reviewer, which have strengthened the clarity and confidence of this manuscript.

Reviewer #3 (Remarks to the Author):

This Reviewer appreciate the efforts to respond to the questions raised during my revision of their work. Most of the issues have been satisfactorily addressed and contribute to a better understanding of the manuscript. The main conclusion of this work is related to the regulatory role of GATA3 enhancer in ILC2 development is well established.

Thank you for evaluating the revised manuscript. We are truly grateful for the insightful comments provided by the Reviewer, which have strengthened the clarity and confidence of this manuscript.

However, the functional data supporting the identification of *Cnot6l* as a mediator of GATA3SE-dependent ILC2 development is not the strongest part of the manuscript as:

- Reconstitution at week 5 does not allow for replenishment of ILC2s in peripheral tissues. Authors should wait a bit longer to analyze mice after BM transplant. Please refer to Jacquelot N., et al 2021.**
- Evidence for efficient downregulation of *Cnot6l* in NGFR1+ cells in vivo is lacking.**
- It is unclear how the quantification of ILC2s in transduced cells is done. Is it from a CD45+ NGFR1+ gate? Have the authors excluded Lineage positive cells or added any other ILC2 marker?**
- Additionally, the authors could have tested the ability of *Cnot6l* overexpression to rescue the *Gata3SEKO* phenotype.**
- In Supp. Fig. 12, shPtpn13 effects on ST2+ cells seem to be comparable to *Cnot6l* downregulation. However, Ptpn13 is regarded as not affecting development of ILC2s. This Reviewer does not understand the criteria for choosing a *gata3SE* candidate based on the data shown. Is there a statistical test done on Supp. Fig 12E?**

In conclusion, I would suggest to include *Cnot6l* functional data (Figure 5G,5h, and 5J) as supplementary information instead. Otherwise, I thank the authors for their work and look forward to see this manuscript published.

We appreciate the Reviewer's suggestion on the ambiguous part related to *Cnot6l* in this manuscript.

As noted by the Reviewer, Jacquelot et al. (2021, Nat Immunol) confirmed that the frequency of circulating eosinophils fully reconstituted at six weeks. However, we analyzed at five weeks as Zhong et al. (2016, Nat Immunol) and Constantinides et al. (2014, Nature), investigated in ILC

development after the adoptive transfer of bone marrow cells at five weeks timepoint. We also avoided longer time points to avoid compensation due to secondary effects, including the over-reconstitution by incompletely knocked-down cells. Therefore, we consider that the analyses at five weeks are reasonable.

Second, unfortunately, we do not have direct evidence for efficient downregulation of *Cnot6l* in $CD45^+hNGFR^+Lin^-Thy1.2^+$ cells in vivo because we could not obtain sufficient cells for checking the expression of *Cnot6l*. It is also possible that sh-*Cnot6l* may lead to the loss of cells dependent on *Cnot6l*, and remaining $CD45^+hNGFR^+Lin^-Thy1.2^+$ cells might express substantial levels of *Cnot6l* due to incomplete suppression by sh-*Cnot6l*. To evaluate the shRNA's knockdown efficacy, we performed a retrovirus-mediated reporter assay that assesses the degree of suppression of *Cnot6l* mRNA (below Figure A).

In the system, two vectors, the sh-vector, which contains hNGFR, and the validation vector, which contains *Cnot6l*-ires-GFP, were simultaneously transduced into cultured Th2 cells, and the degradation of *Cnot6l* mRNA derived from the validation vector was evaluated by the expression of GFP by FACS. The knockdown efficiency was calculated as Integrated MFI (%GFP multiplied by MFI of GFP⁺ population) of hNGFR⁺ cells with the normalization of Integrated MFI of hNGFR⁻ cells. This analysis showed the highest knockdown efficiency of the sh1 vector among three candidate vectors. We, therefore, utilized the sh1 vector for further experiments.

Third, we apologize for the uncertainties in the description of quantifying ILC2s in transduced cells. The shRNA-transduced cells were identified as $CD45^+hNGFR^+$ cells in the lung. We then plotted Lineage vs. Thy1.2 in this population and identified ILCs as $Lin^-Thy1.2^+$ cells. This gating strategy is similar to Supplementary Fig. 4a, using $CD45^+hNGFR^+$ cells instead of $CD45^+$ cells. We described this gating strategy in the revised Figure Legend as below:

"G, Experimental procedure for shRNA-transduced BM chimeric mice. **Non-silencing or sh-*Cnot6l* retrovirus-transduced WT BM cells were transferred into irradiated recipient mice. Five weeks later, lung cells and BM cells were harvested, and shRNA-transduced cells were analyzed by gating hNGFR⁺ cells. H, Lung ILC frequency ($CD45^+hNGFR^+Lin^-Thy1.2^+$ cells) in $CD45^+hNGFR^+$ cells. n=6. I, BM ILC frequency ($hNGFR^+Lin^-CD127^+CD135^-$ cells) in $hNGFR^+$ cells. non-silencing: n=6, sh-*Cnot6l*: n=5."**

Forth, upon another Reviewer's request, we conducted *Cnot6l* overexpression experiments in G3SEKO ILCP using the OP9-DL1 culture system employed for *Gata3* overexpression in Fig. 2d. However, unfortunately, our attempts did not significantly recover $CD25^+PD-1^-$ cell development from *Cnot6l*-overexpressed G3SEKO ILCP (data not shown). These findings may

indicate that *Cnot6l* alone is not a sufficient condition for the impairment of ILC2 development in G3SEKO mice. On the other hand, the results of the new ATAC-seq (Fig. 5e) and luciferase experiment of *Cnot6l*-enhancer (Supplementary Fig. 12g) strengthen the critical role of G3SE in the *Cnot6l* upregulation during ILC2 development.

Fifth, we apologize for the incorrect figures for sh-*Inpp4b* and sh-*Ptpn13* in Supplementary Fig. 12. We could not recognize this mistake until the Reviewer pointed it out. We replaced the figures as shown below. We performed statistical analysis, but there was no significance. We put all p values in the figures according to the Nature Communications format.

As the Reviewer points out, we understand that the *Cnot6l* section is weak in this manuscript. Initially, we considered using Fig. 5h-g as supplementary figures, as recommended by the Reviewer, but after considering the overall structure of this manuscript, we chose to present them as they are, as Fig. 5h-g. Instead, we have added the limitations of our experimental methods and the need for re-evaluation, particularly in the genetically deficient murine models, to the Discussion section.

Page 18, 3rd Paragraph of the Discussion section, added as follows:

"We have identified five candidate genes targeted by GATA3 during ILC2 development. Two of these genes are already known to play pivotal roles in ILC2 development, and we newly proposed a role for *Cnot6l* in this process. The remaining two genes did not significantly affect ILC2 development in our experimental setting; however, given the incomplete suppression in the sh-knockdown experiment, further validation of these genes, including *Cnot6l*, using more robust methodologies, such as genetically deficient mice, is warranted."

Reviewer #4 (Remarks to the Author):

In the revised version of their manuscript, authors Furuya et al. have substantively and significantly improved the quality of their submission. Most importantly, they have added to the significance of the manuscript through the addition of a knockout (G3SE2) that further subdivides their larger G2SE knockout and provides additional insight into the regulation of GATA3 in ILC2s beyond what has been reported in their manuscript and the literature. Furthermore, they have improved the impact of their manuscript through references to published literature, placing their findings in the context of the field at large, and through improvements in experimental rigour, including additional scRNA-seq data and further probing of Cnot6l. Lastly, they have improved the overall clarity of the manuscript through the addition of gating strategies and explanations of ILC2 development, as understood in the field. Overall the revisions the authors added have improved the impact of the paper and will be a welcome addition to the ILC2 and GATA3 field at large. Beyond the additional experiments proposed in the Discussion by the authors, it would be interesting to investigate the chromosomal conformation at the GATA3 locus in ILC2s and developing ILCs, particularly between G3SE and GATA3, and G3SE1 and G3SE2.

Thank you for evaluating the revised manuscript. We are truly grateful for the insightful comments provided by the Reviewer, which have shed light on various aspects that were not apparent to us at the initial submission. We also found the perspective on the changes in chromosomal conformation during ILC2 development, which the Reviewer mentioned, particularly intriguing. We have acknowledged it as a point to further explore in the following steps.